# Start codon variant in *LAG3* is associated with decreased LAG-3 expression and increased risk of autoimmune thyroid disease

Autoimmune thyroid disease (AITD) is a common autoimmune disease. In a GWAS meta-analysis of 110,945 cases and 1,084,290 controls, 290 sequence variants at 225 loci are associated with AITD. Of these variants, 115 are previously unreported. Multiomics analysis yields 235 candidate genes outside the MHC-region and the findings highlight the importance of genes involved in T-cell regulation. A rare 5′-UTR variant (rs781745126-T, MAF = 0.13% in Iceland) in *LAG3* has the largest effect (OR = 3.42, $P = 2.2 \times 10^{-16}$) and generates a novel start codon for an open reading frame upstream of the canonical protein translation initiation site. rs781745126-T reduces mRNA and surface expression of the inhibitory immune checkpoint LAG-3 co-receptor on activated lymphocyte subsets and halves LAG-3 levels in plasma among heterozygotes. All three homozygous carriers of rs781745126-T have AITD, of whom one also has two other T-cell mediated diseases, that is vitiligo and type 1 diabetes. rs781745126-T associates nominally with vitiligo (OR = 5.1, $P = 6.5 \times 10^{-3}$) but not with type 1 diabetes. Thus, the effect of rs781745126-T is akin to drugs that inhibit LAG-3, which unleash immune responses and can have thyroid dysfunction and vitiligo as adverse events. This illustrates how a multiomics approach can reveal potential drug targets and safety concerns.

AITD has a population prevalence of ~5%, most often presenting as hypothyroidism (Hashimoto's thyroiditis) while some cases have hyperthyroidism (Graves' disease), following immune-mediated disturbance of thyroid hormone production[1]. Although AITD is characterized by thyroid autoantibodies, they are also present in ~10% of the population, so only a proportion of individuals who have thyroid autoantibodies, develop a clinically overt AITD[2].

AITD has a strong genetic component[3] and we previously reported a genome-wide association study (GWAS) meta-analysis, identifying 99 sequence variants associated with the disease[4], of which 20 were previously reported[5–10]. Here, we report a meta-analysis of 110,945 cases and 1,084,290 controls, using the same case definition[4,11]. In addition to our previous meta-analysis, AITD diagnoses registered in primary care and by private practitioners in Iceland and UK are now included, as well as study populations from the USA and Finland.

The GWAS was performed with ~56 million variants identified through whole-genome sequencing and imputation into chip-typed individuals. We identified candidate genes for the AITD-associated lead signals through multiomics analyses (Fig. 1). rs781745126-T, a 5′-UTR variant in LAG3, confers largest risk of AITD, we therefore explored its clinical impact as well as its effects on mRNA expression, protein translation and T-cell and B-cell function. Finally, we evaluated the effects of AITD-associated variants that result in amino-acid changes on the protein.

✉e-mail: saedis.saevarsdottir@decode.is; kstefans@decode.is

**Fig. 1 | Multiomics approach to identify sequence variants that associate with autoimmune thyroid disease (AITD) and point to candidate causal genes. a** A GWAS meta-analysis on 110,945 cases and 1,084,290 controls from Iceland, Finland, UK and USA was performed to identify sequence variants that associate with AITD. For lead signals, a systematic variant annotation was applied, identifying AITD lead variants or correlated variants ($r^2 > 0.8$) that affect protein *coding* (Supplementary Data 1 and 3), *mRNA expression* (top cis-eQTL or sQTL, Supplementary Data 4–6) or *levels of proteins* in plasma (top cis-pQTL, Supplementary Data 1 and Supplementary Data 7–8). **b** Out of 280 AITD lead variants outside the MHC-region, 141 sequence variants point to candidate genes, of which 25 variants pointing to 26 genes increased the risk of AITD by ≥10%, as shown in the figure, ranked by effect size (odds ratio, OR), calculated using logistic regression analysis assuming a multiplicative model, with two-sided *P*-values adjusted for year of birth, sex and origin (Iceland) or the first principal components (UK, USA), and Bonferroni adjustments for multiple comparisons, see details in Methods. In addition, the variant at the HLA locus with largest effect is shown. All effects are shown for the AITD risk increasing allele based on meta-analysis of study populations from Iceland, Finland, UK and USA. Previously unreported associations (GWAS catalog with $P < 5 \times 10^{-8}$) with AITD or related phenotypes (Graves' disease, Hashimoto's thyroiditis, hypo- or hyperthyroidism) are marked with *.

## Results
### GWAS meta-analysis
We found 290 genome-wide significant associations at 225 loci using an additive model (Supplementary Fig. 1, Supplementary Data 1, and Fig. 1a). Of the 290 associations, 115 had not previously been associated with AITD or related disease phenotypes (Graves' disease, Hashimoto's thyroiditis, hypo- or hyperthyroidism, Supplementary Data 1–2)[4–10,12].

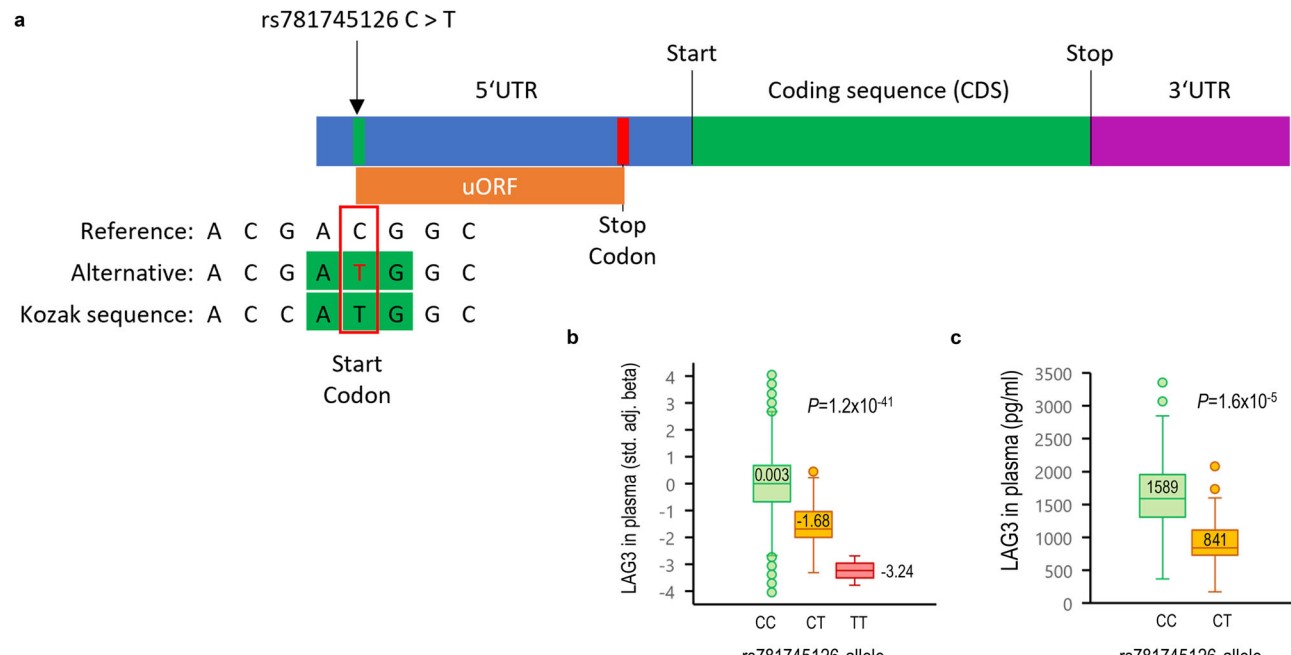

**Fig. 2 | *LAG3* 5' UTR variant rs781745126-T creates a novel upstream open reading frame (uORF) and associates with lower plasma levels of LAG-3.**
**a** rs781745126-T creates a novel start codon, thereby generating a novel upstream open reading frame (uORF), that contains a STOP codon after 84 codons or 51 bp upstream of LAG-3 protein translation initiation site. rs781745126-T generates a stronger Kozak sequence than the reference *LAG3* start codon has, which might reduce the use of the canonical start site, that could result in reduced levels of the LAG-3 protein. **b** Plasma levels of LAG-3 were measured using a proteome-wide screening with the aptamer-based SomaScan® platform in 37,943 Icelanders. rs781745126-T had the largest effect on LAG-3 levels (top cis-pQTL). The distribution of standardized beta adjusted values in two homozygote rs781745126-T

carriers (TT), 93 heterozygote carriers (CT), and 37,848 non-carriers (CC) is shown by box-plots (outliers, $10^{th}$-$90^{th}$ percentile, interquartile range, median levels) and statistical comparison is calculated using linear regression of log-transformed protein levels against SNP allele count (see also Table 1). **c** Lower plasma levels in 66 rs781745126-T carriers (CT, median 841 pg/ml) than in 66 age and sex-matched non-carriers (CC, median 1589 pg/ml) were confirmed in Icelanders by another method, the antibody-based MSD assay (R-PLEX # F213Y-3, Meso Scale Diagnostics) and the distribution is shown as in figure **b**. The correlation (Spearman) between the two methods was high ($R = 0.93$, $P < 2.2 \times 10^{-16}$). All *P*-values are two-sided.

## Multiomics analysis of lead signals

Through a systematic annotation of the 280 AITD lead or correlated variants ($r^2 > 0.8$) outside the major histocompatibility complex (MHC)-region, we found evidence for how 141 of those variants may confer disease risk based on their effects on protein coding, mRNA expression (eQTL), splicing (sQTL), and/or protein levels (cis-pQTL), pointing to 235 candidate genes, as summarized in Fig. 1b and Supplementary Data 1–8.

We identified several sequence variants with a strong effect on AITD risk (Fig. 1b). Two of these are in the same gene, *LAG3*; a rare 5'-UTR variant (rs781745126-T, OR = 3.42, $P = 2.2 \times 10^{-16}$) and a missense variant (rs149722682-A, Pro67Thr, OR = 2.11, $P = 2.2 \times 10^{-9}$). These two sequence variants in *LAG3* illustrate a founder effect in Iceland (rs781745126-T, minor allele frequency (MAF) = 0.13% in Iceland, 0.00016% in UK and no carriers found in other study populations) and in Finland[13] (rs149722682-A, MAF = 0.10%, no carriers found outside Finland). A missense variant in the *ZAP70* gene (rs145955907-T, Thr155Met) that associates with an increased risk of AITD (OR = 1.27, $P = 1.0 \times 10^{-16}$) is also an example of a founder effect in Finland (MAF = 1.95%), although it is present, but rare, in all the other populations (MAF = 0.01–0.03%, see Supplementary Data 9).

## Clinical impact of the Icelandic *LAG3* rs781745126-T variant

We identified 416 heterozygous and three homozygous carriers of the rs781745126-T variant in Iceland. The homozygous carriers were all alive, had children and were 65, 70, and 85 years old at the time of analysis. Strikingly, all three homozygotes for the rs781745126-T variant have AITD and one also has two other autoimmune diseases, type 1 diabetes and vitiligo. We therefore tested the association of

rs781745126-T with type 1 diabetes and vitiligo, and found a nominally increased risk of vitiligo among rs781745126-T carriers (OR = 5.10, $P = 6.5 \times 10^{-3}$) but no association with type 1 diabetes (OR = 1.80, $P = 0.12$). Further testing of 17 other autoimmune diseases yielded no significant association after multiple testing correction (Supplementary Data 10). Two of the three homozygous carriers of rs781745126-T and 15% of the heterozygous carriers have a history of cancer, but there was no association of rs781745126-T with cancer overall ($N = 49,981$; $P = 0.27$) or any of 31 cancer subtypes or with cancer survival, retrieved from nationwide data (Supplementary Data 11, $P > 0.05$ for all). In an Icelandic cohort with thyroid autoantibody measurements (8,196 individuals measured of whom 4667 have AITD and 2442 had thyroid autoantibodies at that timepoint), we found that among those who had thyroid autoantibodies ($N = 1967$, Supplementary Data 10), the risk of AITD was 63-fold higher in rs781745126-T carriers than in non-carriers ($P = 8.2 \times 10^{-3}$).

## Impact of *LAG3* rs781745126-T on transcription, translation, and protein levels

As illustrated in Fig. 2a, rs781745126-T is in the 5' untranslated region (UTR) of the *LAG3* gene, which is a regulatory region that is transcribed into mRNA but not translated into protein. We found that rs781745126-T generates a novel translation initiation site (TIS) 303 bp upstream of the canonical TIS. The new TIS or start codon has a stronger Kozak sequence, that is a stronger nucleic acid motif for the initiation of protein translation[14]. It is predicted to generate a novel upstream open reading frame (uORF) that contains a STOP codon after 84 codons or 51 bp upstream of the canonical LAG-3 protein TIS. Such uORFs have been shown to reduce translation of the downstream ORF and, to a

**Table 1 | Sequence variants at the LAG3 locus that associated with LAG-3 protein levels in plasma (top cis-pQTL) and their association with risk of autoimmune thyroid disease**

| Chrom | Pos | rs | Variant effect | EA | OA | SomaScan (Iceland)* | | | AITD meta-analysis** | | |
|---|---|---|---|---|---|---|---|---|---|---|---|
| | | | | | | 37,943 individuals | | | 110,945 cases; 1,084,290 controls | | |
| | | | | | | MAF (%) | effect in SD (adj.)*** | $P$ | MAF (%) | OR | $P$ |
| chr12 | 6772551 | rs781745126 | 5'-UTR | T | C | 0.13 | −1.62 | $3 \times 10^{-44}$ | 0.07 | 3.42 | $2 \times 10^{-16}$ |
| chr12 | 6773332 | rs149722682 | Pro67Thr | A | C | 0 | – | – | 0.10 | 2.11 | $2 \times 10^{-9}$ |
| chr12 | 6774733 | rs147749635 | Pro217Leu | T | C | 0.13 | −1.31 | $1 \times 10^{-30}$ | 0.06 | 1.27 | 0.03 |
| chr12 | 6771308 | rs7488113 | upstream | T | G | 2.48 | −0.17 | $1 \times 10^{-9}$ | 3.80 | 0.98 | 0.07 |
| chr12 | 6775910 | rs3782735 | downstream | G | A | 38.27 | −0.13 | $1 \times 10^{-51}$ | 40.95 | 0.99 | 0.28 |
| chr12 | 6771830 | rs2301333 | upstream | A | G | 5.94 | 0.13 | $1 \times 10^{-13}$ | 5.72 | 0.98 | 0.12 |

*The Somascan dataset is an Icelandic cohort of cases and controls (irrespective of diseases) that had available plasma samples ($N = 39,155$) and of those, 37,943 also had genotypes and overlaps with the Icelandic cohort of AITD cases and controls (see further description in ref. Ferkingstad, E. et al. Large-scale integration of the plasma proteome with genetics and disease, Nat Genet 2021).
**GWAS meta-analysis of study populations from Iceland (18,651 cases and 326,068 controls), UK (33,822 cases and 396,901 controls), Finland (Finngen release 8: 36,321 cases and 306,178 controls) and USA (22,151 cases and 55,143 controls), in total 110,945 cases and 1,084,290 controls. We used logistic regression analysis assuming a multiplicative model, reporting odds ratios (OR) and two-sided $P$-values adjusted for year of birth, sex and origin (Iceland) or the first principal components (UK, USA) and Bonferroni adjustments for multiple comparisons, see further in Supplementary Data 1 and 9 and Methods. rs781745126 is only present in Iceland and rs149722682 in Finland.
***Effect is adjusted for the lead signal at the locus.

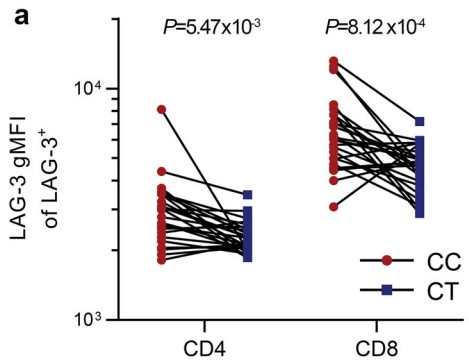

**Fig. 3 | rs781745126-T carriers have reduced LAG-3 expression on surface of activated CD4⁺ and CD8⁺ T-cells.** Peripheral blood mononuclear cells (PBMCs) from 25 heterozygous rs781745126-T carriers (CT, blue) and 25 non-carrier controls (CC, red) matched for age and sex were stimulated with anti-CD3/anti-CD28 beads for 6 days to induce LAG-3 expression and proliferation. **a** LAG-3 surface expression on CD4⁺ and CD8⁺ T-cells (geometric mean fluorescence intensity, gMFI) and (**b**) frequency of proliferating cells analyzed on day 6, using cell trace violet (CTV) stain. Paired t-test with two-sided $P$-values was used to compare carriers (blue) and non-carriers (red).

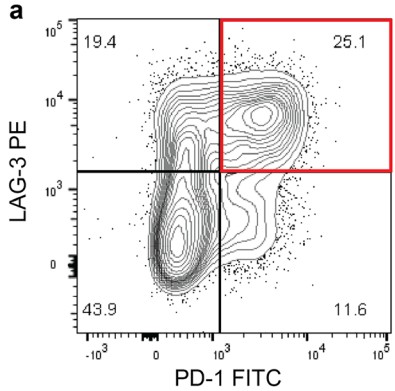
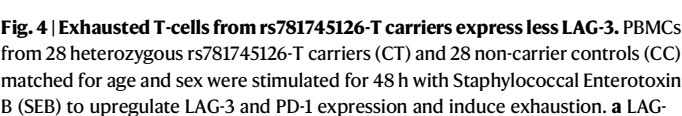
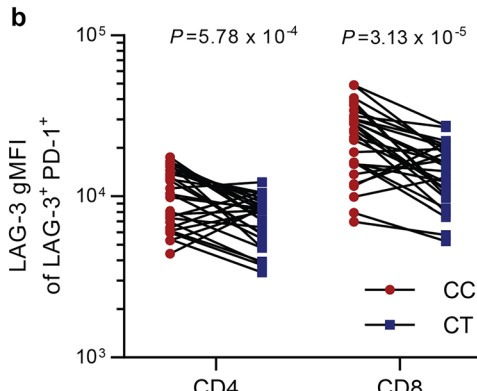

**Fig. 4 | Exhausted T-cells from rs781745126-T carriers express less LAG-3.** PBMCs from 28 heterozygous rs781745126-T carriers (CT) and 28 non-carrier controls (CC) matched for age and sex were stimulated for 48 h with Staphylococcal Enterotoxin B (SEB) to upregulate LAG-3 and PD-1 expression and induce exhaustion. **a** LAG-3⁺PD-1⁺ cells (red square) representing exhausted T-cells. **b** Surface expression intensity (geometric mean) of LAG-3 on double-positive LAG-3⁺PD-1⁺ exhausted CD4⁺ and CD8⁺ T-cells. Paired t-test with two-sided $P$-values was used to compare carriers (blue) and non-carriers (red).

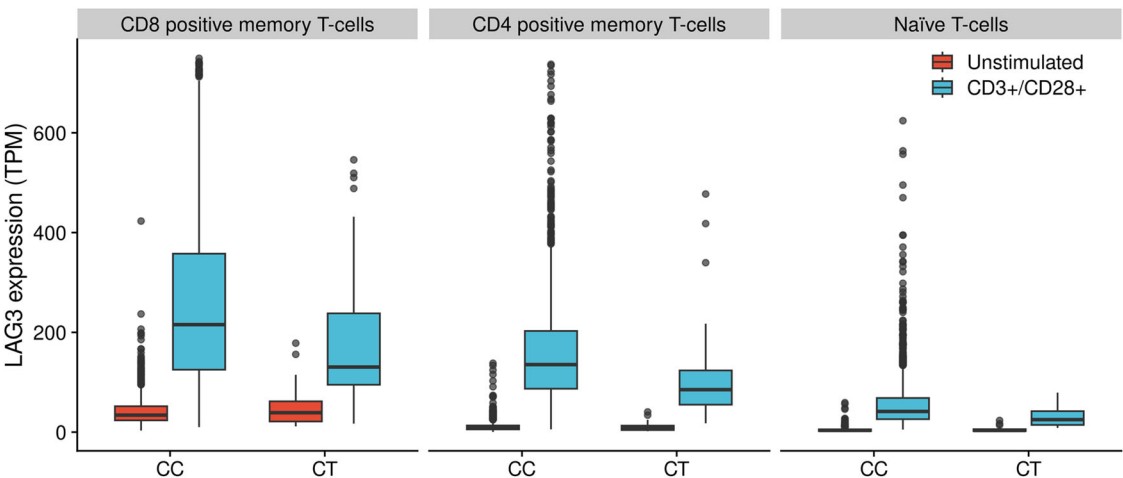

**Fig. 5 | rs781745126-T carriers have less *LAG3* mRNA expression in stimulated T-cell subsets.** LAG3 expression was assessed in data from single-cell RNA (scRNA) sequencing (in-house data) of unstimulated (red) and stimulated (blue) peripheral mononuclear cells (PBMCs, 24-h stimulation with anti-CD3/anti-CD28) from 26 heterozygous carriers (CT) and 766 non-carriers (CC) of rs781745126-T. The y-axis shows the expression in TPM (Transcripts Per Million) and the distribution is shown by box-plots (outliers, 10–90th percentile, interquartile range, median levels). LAG3 expression was increased after stimulation, but this upregulation was lower among carriers of rs781745126-T than among non-carriers on CD8$^+$ T-cells (43% lower, $P = 1.06 \times 10^{-5}$), CD4$^+$ memory T-cells (35% lower, $P = 8.59 \times 10^{-4}$) and naïve T-cells (45% lower, $P = 3.36 \times 10^{-6}$). For all subsets and information about marker genes and cell type classification of scRNA-sequencing data, see Supplementary Data 12, Supplementary Data 13, Supplementary Information and "Methods" section. Two-sided *P*-values were computed using Satterthwaite's method for mixed-effect models.

lesser degree, destabilize the mRNA[15,16]. rs781745126-T may, therefore, reduce the use of the canonical start site, which could result in lower levels of LAG-3.

Indeed, when analyzing the effect of the rs781745126-T variant on levels of 4,907 proteins in plasma (measured in 37,943 Icelanders using the SomaScan platform[17]), we found that the rs781745126-T carriers have lower plasma levels of LAG-3 (Fig. 2b, cis-pQTL effect = -1.62 SD, $P = 2.69 \times 10^{-44}$, Table 1). rs781745126-T had the largest effect on LAG-3 levels of all sequence variants, followed by a rare *LAG3* missense variant (rs147749635-T, Pro217Leu, MAF = 0.13%, effect adjusted for rs781745126-T = −1.31 SD, adjusted $P = 1 \times 10^{-30}$) that associated nominally with AITD risk (OR = 1.27, $P = 0.03$). Three more common sequence variants had independent association with LAG-3 levels, but with ten times less effect and no association with AITD. The effect of rs781745126-T on LAG-3 plasma levels was confirmed with an antibody-based assay (*Meso Scale Diagnostics*, Fig. 2c) with heterozygous rs781745126-T carriers (N = 66) having only half the plasma levels of LAG-3 found in sex and age matched controls (N = 66, median 842 vs. 1589 pg/ml, $P = 1.6 \times 10^{-5}$). rs781745126-T is not the top trans-pQTL of any of the other 4,906 proteins measured in Icelanders.

Next, we evaluated if rs781745126-T affects LAG-3 levels on lymphocyte subsets. T-cells were activated in vitro to induce LAG-3 surface expression (Fig. 3a), and we found that rs781745126-T heterozygous carriers (n = 25) had lower surface expression of LAG-3 than matched non-carriers (n = 25) on both CD4$^+$ ($P = 5.47 \times 10^{-3}$) and CD8$^+$ ($P = 8.12 \times 10^{-4}$) T-cells, without detectable effect on their proliferation (Fig. 3b). Immortalized B-cells from rs781745126-T carriers also expressed less LAG-3 on their surface ($P = 6.70 \times 10^{-3}$) and in the cell supernatant ($P = 2.73 \times 10^{-3}$) than immortalized B-cells from non-carriers (Supplementary Fig. 2).

Persistent antigen stimulation can induce T-cell exhaustion, a state wherein T-cells have reduced effector capacity and an increased expression of inhibitory co-receptors, including the check-point inhibitors LAG-3 and PD-1[18]. Intra-tumoral T-cells that co-express LAG-3 and PD-1 have been shown to be particularly exhausted[19]. We therefore tested whether rs781745126-T affects LAG3 expression on LAG-3$^+$PD-1$^+$ exhausted T-cells (Fig. 4), and found that the surface expression of LAG-3 was lower in rs781745126-T carriers than in non-carriers on both

the CD4$^+$ ($P = 5.78 \times 10^{-4}$) and CD8$^+$ ($P = 3.13 \times 10^{-5}$) T cell subsets. A similar pattern was observed in all LAG-3$^+$ and in the activated (CD25$^+$) subset (Supplementary Fig. 3). Taken together, carriers of rs781745126-T have lower expression of LAG-3 on activated and exhausted T-cells as well as on immortalized B-cells.

The question remains whether decreased levels of LAG-3 among rs781745126-T carriers result from reduced translation or transcription. In order to explore the latter hypothesis, we examined the impact of rs781745126-T on mRNA expression. rs781745126-T did not associate with total mRNA expression (cis-eQTL) in whole blood (n = 17,848, effect = -0.12 SD, $P = 0.39$, Supplementary Fig. 4). Neither did it associate with *LAG3* mRNA expression in unstimulated peripheral blood mononuclear cells (PBMCs) from single-cell RNA-sequencing data of 26 heterozygous carriers and 766 non-carriers of rs781745126-T (Supplementary Data 12–13). However, as expected from the role of LAG-3, T-cells upregulated *LAG3* expression after T-cell (anti-CD3/anti-CD28) stimulation, but this upregulation was lower among carriers of rs781745126-T than among non-carriers in naïve ($P = 3.36 \times 10^{-6}$), CD8$^+$ ($P = 1.06 \times 10^{-5}$), and CD4$^+$ memory T-cells ($P = 8.59 \times 10^{-4}$, Fig. 5 and Supplementary Data 12). Thus, we demonstrate that rs781745126-T is associated with a reduction in LAG-3 levels, which can be attributed, at least in part, to a decrease in mRNA levels in activated T-cells.

## Variants pointing to candidate causal genes through altered protein coding

The other sequence variant in *LAG3* that is associated with AITD, missense variant rs149722682-A, results in an amino-acid change (p.Pro67Thr) in the D1 domain at the base of a large extra-loop structure that has been shown to be important for MHC-class II binding, and decrease the binding (Supplementary Fig. 5a)[20–22]. This is a proline-rich loop which is not found in CD4[20] and the prolines in the Pro67Thr domain are highly conserved among mammals[20]. The rare Finnish Pro67Thr did not associate with other autoimmune diseases or with cancer, after multiple testing correction (Supplementary Data 10).

In the GWAS meta-analysis, we also found a missense variant in the *ZAP70* gene (rs145955907-T, MAF = 0.01–1.95%) that associates with an increased risk of AITD (OR = 1.27, $P = 1.0 \times 10^{-16}$) and results in an amino-acid change (Thr155Met) in a conserved region of ZAP-70,

where it replaces the hydrophilic threonine with hydrophobic methionine in an alpha helix close to the kinase domain[23] (Supplementary Fig. 5b). When tested in meta-analyses of 19 autoimmune diseases and 34 cancer subtypes in the same four study populations as AITD, we found that Thr155Met also increases the risk of type 1 diabetes (OR = 1.33, $P = 1.9 \times 10^{-4}$), Sjogren's syndrome (OR = 1.39, $P = 7.4 \times 10^{-4}$), inflammatory bowel disease (OR = 1.27, $P = 3.3 \times 10^{-6}$) and the ulcerative colitis subset (OR = 1.31, $P = 3.0 \times 10^{-6}$), but no cancer subtypes, after correction for multiple testing ($P = 0.05/53 = 9.4 \times 10^{-4}$).

Coding variants in genes at other novel loci that encode zinc-finger proteins also increase the risk of AITD; a frameshift variant in *ZNF429* (rs199679715-C, MAF = 0.73-2.05%, OR = 1.14, $P = 6.6 \times 10^{-11}$) and a missense variant in *ZNF800* (rs62621812-A, MAF = 2.17-4.23%, OR = 1.10, $P = 3.0 \times 10^{-11}$) that results in an amino-acid change from proline to serine (Pro103Ser). The *ZNF800* gene is highly constrained for loss-of function variants and Pro103Ser is located in a conserved region, a loop in between two alpha helices (Supplementary Fig. 5c). Pro103Ser in *ZNF800* does not associate with other autoimmune diseases or with cancer overall, but is associated with ovarian cancer (OR = 1.29, $P = 8.1 \times 10^{-5}$) and breast cancer (OR = 1.10, $P = 3.2 \times 10^{-5}$) in the same direction as with AITD and the signals co-localize. *ZNF429* is on the other hand not constrained for loss-of-function variants and the rs199679715-C frameshift results in a change in the coding sequence after 526 amino acids with a replacement of the last 148 amino acids with 166 new amino acids, affecting 5.5 of the 18 zinc-finger motifs (Supplementary Fig. 5d). rs199679715-C does not associate with other autoimmune diseases or with cancer. Both *ZNF429* and *ZNF800* are highly expressed in blood, immune cells, and endocrine glands. Whether these zinc-binding proteins regulate lymphocyte activation through zinc binding, affect transcriptional regulation or other mechanisms that might contribute to the association with AITD, remains to be elucidated.

In addition, we identified several unreported missense variants that confer substantial risk of AITD and point to *IL4R*, *TSHR*, *SH2B3*, *IRF4*, *IRF3*, and other candidate genes (Fig. 1b and Supplementary Data 1).

## Discussion

In this large GWAS and multiomic analysis of AITD, we identified 290 sequence variants that associate with AITD at 225 loci and 115 had not previously been associated with AITD or related disease phenotypes. Multiomics analysis yielded 235 candidate genes outside the MHC-region. The findings highlight the importance of genes involved in T-cell regulation, where a 5′-UTR variant (rs781745126-T) in *LAG3* has the largest effect, with a 3.4-fold increased risk of AITD. *LAG3* encodes lymphocyte activation gene 3 (LAG-3) protein and rs781745126-T generates a novel start codon for an open reading frame upstream (uORF) to the canonical protein coding sequence of *LAG3*. Such uORFs have been shown to reduce translation of the downstream ORF[15,16] and may, therefore, result in lower protein levels. Indeed, we observe that the rs781745126-T variant halves LAG-3 levels in plasma among heterozygotes.

LAG-3 is an inhibitory co-receptor that is reported to be expressed by activated (CD4[+], CD8[+]) and regulatory T-cells in humans[24]. It is structurally homologous to CD4 and binds to MHC-class II molecules with higher affinity than CD4, thereby preventing antigen-mediated signaling[25]. Recent findings also suggest that LAG-3 could inhibit T-cell receptor signaling in the absence of MHC-class II, which could at least partly explain the inhibitory role LAG-3 has on MHC-class I-restricted CD8[+] T-cells[24,26]. Prolonged exposure to antigen and inflammatory signals increases LAG-3 expression and leads to a state of exhaustion in T-cells, resulting in a reduced capacity to kill viruses and malignant cells, while inhibition of LAG-3 enhances viral control and anti-tumor immune responses[27,28]. A proteolytic cleavage of surface LAG-3 creates soluble LAG-3 in serum and plasma, but the functional role of this soluble form is not completely understood[29,30]. Our results are consistent with the fact that reduced LAG-3-mediated T-cell regulation predisposes to AITD.

We demonstrate with single-cell RNA sequencing that the reduction in LAG-3 levels can be attributed, at least in part, to a decrease in LAG3 mRNA levels. Thus, T-cells upregulated *LAG3* expression after stimulation (anti-CD3/anti-CD28), but this upregulation was lower among carriers of rs781745126-T than among non-carriers both in naïve, CD8[+] and CD4[+] memory T-cells. In line with that, LAG-3 expression was lower among rs781745126-T than non-carriers on both activated and exhausted T-cells as well as immortalized B-cells, both on their surface and in the cell supernatant.

There are currently several drugs inhibiting LAG-3 registered or in clinical trials to treat cancer[31]. However, unlike its precursor immune checkpoint inhibitor drugs targeting CTLA-4, PD-1 and PD-L1[32], LAG-3 drug development has hitherto not had any genetic support for disease association. CTLA-4 agonist (abatacept) is used to treat autoimmune diseases[25] and common sequence variants at the loci of the *CTLA4* and *CD274* genes, encoding the CTLA-4 and PD-L1 checkpoint proteins, are previously reported to associate with AITD and related phenotypes, and confirmed in our meta-analysis (rs911760 and rs11571297, see Supplementary Data 1). Although there are two independent signals at the *CTLA4* locus in our current meta-analysis, they did not fulfill the criteria in the multiomics analysis for *CTLA4* being a candidate causal gene. Nevertheless, the whole-genome sequencing data available in our study populations (Methods) show that the lead signals associate with structural variants in *CTLA4*, thereby pointing to *CTLA4* as a candidate gene in AITD, where rs11571297 represents a common 42 base-pair deletion in the 3′ untranslated region (UTR) of *CTLA4*, while there are several less frequent deletions of variable length. This is in line with a previous report of variations in the length of dinucleotide (AT) repeats within the 3′UTR of *CTLA4*, where repeat length correlated with mRNA and protein levels of CTLA-4[33]. Our observation that all three homozygotes for the *LAG3* rs781745126-T variant have AITD indicates a high penetrance of the variant. Interestingly, one of the homozygotes also has two other T-cell mediated autoimmune diseases, type 1 diabetes and vitiligo, that like thyroid dysfunction, are known immune-related side effects of anti-LAG-3 and the other checkpoint inhibitors[32,34–36]. rs781745126-T confers a fivefold increased risk of vitiligo, but neither of the *LAG3* variants associated with other autoimmune diseases or with cancer after multiple testing correction, although the power to detect associations of lower magnitude or in less common diseases is limited for such rare variants.

A substantial proportion of patients receiving immune checkpoint inhibitor therapy experience thyroid dysfunction, but it is interesting to note that immune-related adverse events have been associated with longer survival in treated patients[37,38]. Furthermore, patients receiving PD-1 or PD-L1 inhibitors that have high polygenic risk score for hypothyroidism are reported to have longer survival but also a higher risk of hypo- and hyperthyroidism as well as vitiligo[39]. In the Icelandic study population, information about survival is available for 9 cancer subtypes, but rs781745126-T did not associate with survival after multiple testing correction (Supplementary Data 11).

Although it may not be practical to test rare variants as clinical predictors of immune-related adverse events in patients receiving anti-LAG3 inhibitors, our findings provide important evidence for the potential adverse events on LAG-3 inhibiting therapy, since the effect of rs781745126-T among carriers is similar to the effect of LAG-3 inhibiting therapy, and illustrates the power of a multiomics approach to reveal not only potential drug targets but also safety concerns.

Only a subset of individuals that have thyroid autoantibodies develop clinically overt AITD[2], but we found that in individuals tested for thyroid autoantibodies on clinical indications, that the risk of AITD was 63-fold higher in rs781745126-T carriers than in non-carriers,

providing further indication that the effect of the rs781745126-T variant is akin to the drugs that inhibit LAG-3, which unleash immune responses and commonly cause thyroid dysfunction. One limitation of our study is that only a minority of the AITD cases had available thyroid autoantibody measurement in the nationwide database, which is available since 2005. Since there are no validated diagnostic criteria for AITD or its subsets, and clinical recommendations do not recommend testing of thyroid autoantibodies on a routine bases in hypothyroidism, as it does not change the therapy[11], it is likely that only a subset of the cases has been tested in routine care.

Compared to our previously reported meta-analysis of 30 thousand AITD cases and 725 thousand controls in study populations from Iceland and UK[4], diagnoses registered in primary care and by private practitioners were now available in Iceland (as well as in UK, the USA, and Finland) which resulted in almost four times larger study population of AITD, both overall and the Icelandic subset. A low-frequency stop mutation in *FLT3* (rs76428106-C) had the strongest effect on AITD risk in our previous report[4] and we validate this association here (OR = 1.34, $P = 2.30 \times 10^{-44}$). We did not have the power to detect the association of the Icelandic rs7817451126-T variant with AITD in our previous report, while it reached the genome-wide significance level in the current study, highlighting the importance of large study populations to detect associations with rare variants, especially those that are enriched in certain populations. The rs781745126-T and Pro67Thr variants in *LAG3* that associate with an increased risk of AITD in Iceland and Finland, respectively, exemplify a founder effect in these populations, that have been rather isolated for centuries. Rare variants may not explain a large proportion of AITD cases in the overall population, but the risk is substantial among carriers and these variants point to *LAG3* as a candidate gene through reduced transcription and protein levels on one hand, and altered protein structure on the other hand, indicating that LAG-3 reduction predisposes to AITD. This demonstrates the power of bottlenecked populations to identify rare disease-associated variants with high risk, that provide insight into the pathogenesis and potential identification of drug targets[13,40].

Another example of population enrichment of rare variants is *ZAP70* Thr155Met, which has a minor allele frequency of 1.95% in Finland and only 0.01-0.03% in Iceland, UK and USA, resulting in higher statistical power to detect an association with AITD in Finland than in the other populations. *ZAP70* encodes ZAP-70, a cytoplasmic tyrosine kinase acting downstream of the T-cell receptor, another important regulator of T-cell activation. In line with our finding, deficiency in T-cell receptor signaling due to *ZAP70* mutations has been reported to result in a more self-reactive T-cell receptor repertoire[41], and autosomal recessive forms of ZAP-70 deficiency result in a combined immunodeficiency that can manifest as autoimmunity and recurrent infections[42].

To conclude, we have, through a multiomics analysis, identified many candidate genes in AITD and illustrated how strong signals pointing to the *LAG3* gene have a founder effect in Iceland and Finland. The Icelandic *LAG3* variant generates a novel start codon for protein coding and results in a reduced capacity to induce mRNA expression of LAG3 in T-cell subsets. Both activated and exhausted T-cells as well as immortalized B-cells have lower expression of LAG-3 on their surface and in cell supernatant. The carriers have only half the plasma levels of LAG-3 compared with non-carriers and all three homozygous carriers have AITD. Other variants in the *LAG3, ZAP70* and other genes, which result in amino-acid changes at conserved protein regions and are likely alter the conformation, call for further investigation. This illustrates how a multiomics approach can in a hypothesis-free manner reveal potential drug targets and their biological effects, where the effect of the *LAG3* start codon variant is comparable to the drugs that inhibit LAG-3 that can induce thyroid dysfunction.

## Methods

### Study Populations for genome-wide association study meta-analysis

The primary GWAS was performed among 110,945 cases with autoimmune thyroid disease (AITD) and 1,084,290 controls from four cohorts; Iceland (deCODE genetics), UK (UK Biobank), USA (Intermountain) and Finland (FinnGen, r8). Compared with our previous meta-analysis[4], with 30 thousand cases from Iceland and the UK Biobank, we have added AITD diagnoses registered in primary care and by private practitioners in Iceland and UK, as well as study populations from the USA and Finland, using the same case definition (see detailed description below, of autoimmune thyroid diagnoses and thyroid hormone prescriptions with exclusion of non-autoimmune causes of thyroid dysfunction (thyroid cancer/surgery, drug-induced and iodine-deficiency)[4,11].

*In Iceland*, a large fraction of the population of 360 thousand inhabitants has participated in the nationwide research at deCODE genetics. Genotyping of the Icelandic study population is based on whole-genome sequence data from the whole blood of 63,460 Icelanders participating in various disease projects at deCODE genetics and imputation into chip-typed individuals and familial imputation into non-chip-typed Icelanders[43,44]. Including the sequenced individuals, 173,025 Icelanders have been genotyped using Illumina SNP chips. Icelandic genealogy was used to calculate genotype probabilities for untyped relatives. The sequencing was done using Illumina standard TruSeq methodology. Only samples with a genome-wide average coverage of 20X were considered. Autosomal SNPs and INDEL's were called using Graphtyper version 1.4[45]. Variants that did not pass quality control were excluded from the analysis. Information about haplotype sharing was used to improve variant genotyping, taking advantage of the fact that all sequenced individuals had also been chip-typed and long-range phased[46].

This resulted in a study population of 346,753 individuals, including cases with AITD, other autoimmune diseases, or with cancer, as described below, identified at Landspitali, the National University Hospital of Iceland (the only tertiary care hospital in Iceland), and from the Registers of Primary Health Care Contacts and of Contacts with Medical Specialists in Private Practice, both kept by the Directorate of Health. Data from Landspitali were collected from years 1977–2019 and from the Directorate of Health between 2010–2019. This includes measurement of thyroid autoantibodies from the only department of clinical immunology in Iceland, available from 2005. Information about cancer diagnoses were retrieved from the nationwide Icelandic Cancer Registry. All participating individuals who donated blood signed informed consent. The personal identities of participants were encrypted using a third-party system approved and monitored by the Icelandic Data Protection Authority[47]. The study was approved by the National Bioethics Committee (approval no. VSN-16-042, VSN 17-171, VSN 18-115) following evaluation of the Icelandic Data Protection Authority.

*The UK Biobank* is a large prospective cohort study in the UK established by the Medical Research Council (MRC) and the Wellcome Trust to study the causes of major adult-onset diseases. In addition to phenotypic data, the participants (500,000) have been genotyped using Axiom(R) Arrays[48]. Of those, 431,079 were genotypically verified of white British (Caucasian) origin and serve as basis for the current study[49]. Variants imputed into UK Biobank samples were derived from whole-genome sequencing of 131,958 UK individuals, performed jointly by deCODE genetics and the Wellcome Trust Sanger Institute[49], using Graphtyper[45] to identify over 245 million high-quality sequence variants and indels. Quality-controlled chip genotype data were phased using SHAPEIT 4[50]. This research has been conducted using the UK Biobank Resource under Application Number "56270".

*In the USA*, Intermountain Healthcare is a Utah-based healthcare system of 24 hospitals and 160 clinics. In a collaboration project,

samples collected by Intermountain have been genotyped at deCODE genetics, using the Infinium Global Screening Array (GSA) from Illumina. A subset of 16,661 individuals were whole-genome sequenced using Illumina technology, and joint variants were called using Graphtyper[45]. A haplotype reference panel was created using the sequence variants and the long-range phased chip data, and used to impute into chip-typed samples[43,44]. The imputation dataset included 79,085 samples identified to be of Caucasian origin using ancestry analysis with ADMIXTURE[51], run in supervised mode using the 1000 Genomes populations CEU, CHB, ITU, PEL, and YRI as training samples. This study has been approved by the Intermountain Healthcare Institutional Review Board, and all participants have provided written informed consent. The data is sourced from Intermountain INSPIRE Registry of individuals with heart disease and HerediGene, a general population study. There was no heterogeneity observed in the GWAS findings between the two Intermountain subcohorts.

*In Finland*, the FinnGen research project, controlled by the University of Helsinki, has provided publicly available GWAS results for numerous phenotypes through the online FinnGen database. The study collected samples from biobanks in Finland and phenotype data at national health registries. All participants have provided written informed consent, and the study has been approved by the Coordinating Ethics Committee of Helsinki and Uusimaa Hospital District. For information on genotyping in FinnGen, see online documentation: https://finngen.gitbook.io/documentation/methods/genotype-imputation.

## Case definitions

**Autoimmune thyroid disease.** Individuals who had received a diagnosis of Graves' disease (E05.9) or Hashimoto's thyroiditis (E06.3) were considered AITD cases as well as those who had been diagnosed with other hypothyroidism (E03.9) and/or had received thyroxin-treatment (ATC-code H03AA01), excluding known non-autoimmune causes of hypothyroidism (thyroid cancer, drug-induced hypothyroidism (E03.2 or ATC-drug codes for lithium (N05AN01), amiodarone (C01BD01) and interferon (L03AB) treatments)).

The number of AITD cases and controls in the study populations were as follows: Iceland (18,651 cases and 326,068 controls), UK Biobank (33,822 cases and 396,901 controls), Intermountain (22,151 cases and 55,143 controls) and Finngen release 8 (36,321 cases (see URL https://r8.risteys.finngen.fi/phenocode/E4_HYTHY_AI_STRICT) and 306,178 controls); in total 110,945 cases and 1,084,290 controls.

**Secondary disease phenotypes.** We tested the lead signals in AITD in other autoimmune diseases if the total number of cases was over 500 cases in meta-analyses of the same study populations (or 100 cases in the Icelandic or Finnish cohorts, for the population-specific variants). Cases were defined by clinical diagnoses and/or ICD10 codes: type 1 diabetes (E10), celiac disease (K900), systemic lupus erythematosus (M329), rheumatoid arthritis (M058, M059, M060, M068, M069) or it's seropositive (M058, M059) and seronegative (M060, M068) subsets (defined by ICD10 codes or by positivity for rheumatoid factor and/or anti-CCP antibodies, as previously described)[52]; multiple sclerosis (G35), ankylosing spondylitis (M45), Sjögren's syndrome (M350), inflammatory bowel disease (K50, K51) and it's subsets ulcerative colitis (K51) and Crohn's disease (K50), psoriasis (L40), psoriatic arthritis (L405/M073) and primary biliary cirrhosis (K473), vitiligo (L12) and myasthenia gravis (G70).

**Malignancies.** Diagnoses of malignancies, including hematological and thyroid (thyroid cancer is excluded from the AITD phenotype), were retrieved from the cancer registries[53] and databases in the study populations collecting information based on the ICD system and includes information on histology (systemized nomenclature of medicine, SNOMED) and for a subset of solid cancers in the Icelandic

cohort, time from diagnosis to death is also available. The study complies with all relevant ethical regulations.

## Statistical analyses

**Association testing and meta-analysis.** We used logistic regression analysis assuming a multiplicative model to test for association between AITD and sequence variants in each of the three cohorts (Iceland, UK, and USA), adjusting for year of birth and sex and also the origin (Iceland) or the first principal components that described significant population structure according to their eigenvalues and loadings (UK $N = 20$, Intermountain $N = 4$), as previously described[54–56]. The association analysis of the imported Finnish data was adjusted for sex, age, the genotyping batch, and the first 10 PCs. We used LD score regression intercepts[57] to adjust the χ2 statistics and avoid inflation due to cryptic relatedness and stratification, using a set of 1.1 million variants. $P$-values were calculated from the adjusted χ2 results. All statistical tests were two-sided unless otherwise indicated.

For the meta-analyses, we combined GWASs from Iceland, UK, and USA with summary statistics from Finland, using a fixed-effects inverse variance method based on effect estimates and standard errors in which each dataset was assumed to have a common OR, but allowed to have different population frequencies for alleles and genotypes. Sequence variants were mapped to NCBI Build38 and matched on position and alleles to harmonize the datasets. The total number of variants included in the meta-analyses was about 56 million variants with MAF > 0.0001. In a random-effects method, a likelihood ratio test was performed in all genome-wide associations to test the heterogeneity of the effect estimate in the four datasets; the null hypothesis is that the effects are the same in all datasets and the alternative hypothesis is that the effects differ between datasets.

As the probability of a variant affecting a phenotype differs between variant functional annotation classes, we applied genome-wide significance thresholds using a weighted Bonferroni procedure that takes into account the prior probability[58]. Thus, sequence variants were split into five classes based on their genome annotation, and significance threshold for each class was based on the number of variants in that class (e.g. lower thresholds for loss of function (high impact) and missense variants (moderate impact)), as previously described[58]. The adjusted significance thresholds are $1.95 \times 10^{-7}$ for variants with high impact, $3.91 \times 10^{-8}$ for variants with moderate impact, $3.55 \times 10^{-9}$ for low-impact variants, $1.78 \times 10^{-9}$ for other low-impact variants in DNase I hypersensitivity sites and $5.92 \times 10^{-10}$ for all other variants including those in intergenic regions.

The primary signal at each genomic locus was defined as the sequence variant with the lowest Bonferroni adjusted $P$-value using the adjusted significance thresholds described above and the results are presented in Supplementary Data 1. Conditional analysis was performed to identify possible secondary signals within 500 kB from index variants. This was done using genotype data for the Icelandic, UK and USA datasets and an approximate conditional analysis implemented in the GCTA software[59] for the Finnish summary data. We excluded the Finnish data from the conditional analysis of the MHC region as the approximate conditional analysis was not reliable due to the high number of independent signals and complex LD structure in that region. $P$-values and odds ratios were combined using a fixed-effects inverse variance method. Class-specific genome-wide significance threshold were also used for the secondary signals.

Novelty of the GWAS significant loci was evaluated based on the GWAS catalog (https://www.ebi.ac.uk/gwas/home), with significance threshold set at $5 \times 10^{-8}$ and searching for all variants correlated ($R^2 > 0.8$) with the primary and secondary signals with AITD or related disease phenotypes (Graves' disease, Hashimoto's thyroiditis, hypo- or hyperthyroidism), see summary in Supplementary Data 1 and Supplementary Data 2.

## Functional evaluation of identified sequence variants

We performed a systematic variant annotation of the AITD associations to identify candidate genes, through identification of coding variants and variants affecting mRNA expression, as described below. Candidate gene refers to the gene that mediates the effect on the disease at a given locus with the highest probability, based on either the variants or highly correlated variants ($R^2 > 0.8$) being coding in the candidate gene or having the strongest and significant effect on mRNA expression of the candidate genes (top cis-eQTL).

**RNA sequencing.** RNA sequencing analysis was performed on whole blood ($N = 17,848$) and adipose tissue ($N = 700$) from Icelanders. RNA isolation was performed using RNAzol RT according to manufacturer's protocol (Molecular Research Center #RN 190). RNA was isolated using Qiagen RNA midi kit according to the manufacturer's instructions. Concentration and quality of the RNA was determined with an Agilent 2100 Bioanalyzer (Agilent Technologies). RNA was prepared and sequenced on the Illumina HiSeq 2500 and Illumina Novaseq systems according to the manufacturer's recommendation.

Gene expression was computed based on personalized transcript abundances estimated using kallisto[60]. Association between sequence variants and gene expression (cis-eQTL) was estimated using a generalized linear regression, assuming additive genetic effect and quantile-normalized gene expression estimates, adjusting for measurements of sequencing artefacts, demography variables, and leave-one-chromosome-out principal components of the gene-expression matrix[57]. We tested whether lead sequence variants associated with AITD were in strong linkage disequilibrium ($R^2 > 0.8$) with top cis-eQTL variants for genes expressed in whole blood ($N = 17,848$) and in adipose tissue ($N = 700$). In addition, we identified top cis-eQTL and sQTL variants for gene expression in several tissue types in GTEx (https://gtexportal.org) and published accessible data (see overview in Supplementary Data 4, results in Supplementary Data 5-6).

## Plasma protein levels and association with sequence variants

**Iceland.** Blood was collected in EDTA tubes that were inverted 4–5 times and then centrifuged for 10 min at $3000 \times g$ at 4 °C. Plasma samples were frozen in aliquots at −80 °C. Plasma aliquots were allowed to thaw on ice and kept away from light during defrosting. Before measurement, the aliquots were mixed by inverting the tubes a couple of times and then centrifuged for 10 min at $3220 \times g$ at 4 °C. We measured 4,907 proteins in plasma samples from 37,943 Icelanders with genetic information and biological samples available as deCODE genetics, using the SomaScan® (*SomaLogic Inc.*), as previously described[17]. The staff running proteomics assays was blinded for samples' genotypical information. Details of the method have been published elsewhere[61,62]. In short, aptamers were modified to recognize specific proteins and an assay quantifies them on a DNA microarray after the protein concentrations are converted into DNA aptamer concentrations. We performed a proteome-wide association study and evaluate whether AITD-associating sequence variants were associated with protein levels (pQTL). Statistical analysis of the findings was performed using linear regression of log-transformed protein levels against SNP allele count, and the significance threshold was corrected for multiple testing of the 4907 proteins ($P < 0.05/4907 = 1.0 \times 10^{-5}$).

**UK Biobank.** We also tested whether AITD-associated sequence variants were associated with protein levels (pQTL) on the Olink platform, including 3072 immunoassays, that has been used to measure plasma samples from 47,151 European participants from the UK Biobank.

A total of 1500 of the plasma samples from Icelanders that were measured on the SomaScan were also tested on the Olink platform,

and there is a strong correlation between LAG-3 levels measured on the platforms (R = 0.75) among those. This correlation is among the highest of those 1800 proteins that were measured on both platforms and the cis-pQTLs identified for LAG-3 levels on SomaScan (Table 1) were also cis-pQTLs for LAG3 measured on the Olink platform.

**Cell culture.** PBMCs were isolated from venous blood samples via standard Ficoll-Paque (*GE Health*, #17144002) density gradient centrifugation at 800 G for 15 min in 50 ml Blood-Sep spin tubes (*DACOS*, #037100SI) and cryopreserved in liquid nitrogen. Prior to use cells where thawn and incubated over night at 37 °C and 5% $CO_2$ at $1.5 \times 10^7$ cells/mL in RPMI 1640 supplemented with 10% fetal bovine serum (FBS) and 1x Penicillin-Streptomycin (*Gibco*, #15140148) (cRPMI). After resting overnight cells were filtered, counted and seeded in a 96-well plate at $1 \times 10^6$ cells/well. Cells were either stimulated with 0.5 μg/mL Staphylococcal enterotoxin B (SEB, *Sigma*, #4881) for 2 days in 100 μL cRPMI or stained with 0.5 μM CellTrace Violet (CTV) (*Invitrogen*, #C34557) according to manufacturer's protocol before stimulation with (5 μL) CD3/CD28 Dynabeads (*Gibco*, #11131D). Lymphoblasts were cultured in upright culture flasks in cRPMI.

**Protein structure.** Structural models of proteins are created with the Alphafold software[63] for the reference (wild-type) proteins of LAG-3 (AF-P18627-F1), ZNF800 (AF-Q2TB10-F1) and ZNF429 (AF-Q86V71-F1) and the X-ray structure of ZAP-70 with PDB: 2ozo. The protein illustrations that are shown in Supplementary Fig. 5 are made in the Chimera software[64].

**FACS staining.** Cells were stained in U bottom 96-well plates. Cells were washed in PBS and Fc receptors blocked with TruStain FcX (*Biolegend*, #422302) according to manufacturer's protocol. Live/Dead fixable aqua dead cell stain (*Invitrogen*, #L34957) was added to the Fc block and incubated at RT for 20 min. Cells were washed with FACS buffer (PBS + 2% FBS) and stained for 20 min at RT with the following antibodies were purchased from *Biolegend*: LAG-3 PE (#369306), PD-1 FITC (#329904), CD20 APC-Cy7 (#302314, Lymphoblasts), CD3 APC-Cy7(#300318, PBMC), CD4 BV605 (#300556, PBMC), CD8 PE-Cy7 (#301012, PBMC) and CD25 APC (#302510, PBMC). Events were captured using Attune Nxt FACS analyser (*Thermo Fisher Scientific*) and data analyzed in FlowJo (*BC*, Version 10.7.1). As shown in Supplementary Fig. 6, Cells were first gated for lymphocytes using SSC-A and FSC-A, then single cells were selected using FSC-H and FSC-A, then live CD3+ cells were selected using Live/Dead vs CD3-APC-CY7, CD4+ or CD8+ cells were then selected either by CD4-BV605 vs CD8-PE-Cy7, then either CD25+ or CD25- cells were selected by using CD25-APC vs SSC-A, finally cells were gated using LAG3-PE vs PD1-FITC.

**Soluble LAG-3 MSD.** Blood was collected in EDTA tubes that were inverted 4–5 times and then centrifuged for 10 min at $3000 \times g$ at 4 °C. Plasma samples were frozen in aliquots at −80 °C. Plasma aliquots were allowed to thaw on ice and kept away from light during defrosting. Before measurement, the aliquots were mixed by inverting the tubes a couple of times and then centrifuged for 10 min at $3220 \times g$ at 4 °C. Soluble LAG-3 in plasma and cell medium was then measured by using MSD R-PLEX Human LAG-3 (# F213Y-3) according to manufacturer's protocol (*Meso Scale Diagnostics*).

**Single-cell RNA-sequencing.** 200,000 PBMCs were cultured in 200 μL cRPMI in the presence or absence of 1 μL ImmunoCult Human CD3/CD28 T Cell activator (*STEMCELL Technologies*, #10971) for 24 hours. Eight samples from different individuals were pooled and loaded onto two separate lanes on a Chromium X (10x genomics) at 18,250 cells per individual sample per lane (at a total number of 146,000 cells per lane) using Chromium Next GEM Single Cell 3' HT Kit v3.1 (10x Genomics, #1000348) and Chromium Next GEM Chip G

Single Cell Kit (10x Genomics (10x Genomics, #1000120) following manufacturer's instructions. All procedures for Post GEM-RT Cleanup and cDNA Amplification were according to manufacturer's instructions (Chromium Next GEM Single Cell 3′ HT Reagent Kits v3.1, Dual Index) and are described, with the methods for gene expression and single-cell RNA sequencing analysis in Supplementary Information and Supplementary Data 13.

## Reporting summary

Further information on research design is available in the Nature Portfolio Reporting Summary linked to this article.

## Data availability

The data with sequence variants passing GATK filters in our previously described Icelandic population WGS data have been deposited at the European Variant Archive database under accession number PRJEB15197. The GWAS summary statistics are available in Supplementary Data and at https://www.decode.com/summarydata/. Finn-Gen data are publicly available and were downloaded from https://www.finngen.fi/en/access_results. The UK Biobank data were downloaded under application no. 56270. The meta-analysis association results and other data supporting the findings of this study are available within the article, in Supplementary Data or Source Data. Proteomics data and protein mapping to UniProt identifiers and gene names were provided by SomaLogic and Olink and the results are provided in Supplementary Data. The authors declare that the data supporting the findings of this study are available within the article, in supplementary files or at https://www.decode.com/ Source data are provided with this paper.

## Code availability

We used publicly available codes and software in conjunction with the methods described, that is available under the following URLs: Graphtyper version 2, https://github.com/DecodeGenetics/graphtyper PANTHER v.16.0, http://www.pantherdb.org/tools/ Variant Effect Predictor (release 100), https://github.com/Ensembl/ensembl-vep IMPUTE2 version 2.3.1, https://mathgen.stats.ox.ac.uk/impute/impute_v2.html dbSNP version 140, http://www.ncbi.nlm.nih.gov/SNP/ STAR software package, version 2.7.10, https://github.com/alexdobin/STAR Ensembl version 87, https://www.ensembl.org/index.html LeafCutter version 1, https://github.com/davidaknowles/leafcutter kallisto version 0.46, https://github.com/pachterlab/kallisto Eagle https://alkesgroup.broadinstitute.org/Eagle/ ADMIXTURE v1.23 http://www.genetics.ucla.edu/software PLINK v.190b3a http://pngu.mgh.harvard.edu/purcell/plink/ UMAP https://github.com/diazale/umap_review GORpipe https://github.com/gorpipe/gor UCSC Browser https://genome.ucsc.edu/ COLOC software package https://cran.r-project.org/web/packages/coloc/vignettes/a01_intro.html Alphafold: https://github.com/google-deepmind/alphafold GWAS catalog https://www.ebi.ac.uk/gwas/ We used R, version 3.6.0 to analyze data and create plots: https://www.r-project.org/, https://ggplot2.tidyverse.org/No custom code was written for this study.

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

## Acknowledgements

We thank the individuals who participated in this study and the staff at the deCODE Recruitment Center and the deCODE genetics core facilities. Further thanks to all our colleagues who contributed to the data collection and phenotypic characterization of clinical samples as well as to the genotyping and analysis of the whole-genome association data. Those fulfilling authorship criteria are included as co-authors and has complied with all relevant ethical regulations. This research has been conducted using FinnGen release r8 and UK Biobank Resource (application number 56270) and the study was approved by the National Bioethics Committees in Iceland (approval no. VSN-16-042, VSN 17-171, VSN 18-115) and the Intermountain Healthcare Institutional Review Board (for further details, see Methods).

## Author contributions

S.S., K.B., T.M., K.G., T.A.O., D.F.G., P.M., G.Thorleifsson, G.L.N., I.J., and K.S. designed the study and interpreted the results. S.S., T.M., K.B., K.G., J.G., A.J.J., B.L., G.G., K.U.Knowlton., S.K., L.D.N., H.H., T.R., G.Thorleifsson, I.J., and K.S. carried out the subject ascertainment and recruitment. S.S., K.B., T.M., J.B., G.H.Halldorsson, G.R., B.V.H., K.G., A.O.A., S.H.L., L.S., G.M., G.H. Eldjarn, K.Kristjansdottir, K.H.S.M., S.A.G., S.R., O.T.M., P.S., D.F.G., P.M., G.A.Thorisson, and I.J. performed the sequencing, genotyping, expression and proteomics analyses. S.S., K.B., T.A.O., K.G., A.O.A., P.M., G.L.N., and I.J. planned and performed the functional lab work. S.S., K.B., T.M., J.B., G.H.Halldorsson, S.H.L., L.S., A.S., A.O., E.F., E.V.I., G.S., G.M., P.S., D.F.G., T.R., P.M., G.Thorleifsson, and I.J. performed the statistical and bioinformatics analyses. S.S., K.B., T.M., T.A.O., G.R., K.Kristjansdottir, D.F.G., T.R., P.M., G.Thorleifsson, G.L.N., I.J., and K.S. drafted the manuscript. All authors contributed to the final version of the paper.

## Competing interests

S.S., K.B., T.M., J.B., T.A.O., G.H.H., G.R., K.G., A.O.A., S.H.L., L.S., J.G., A.S., A.O., B.V.H., E.F., E.V.I., G.S., G.M., G.H.E., G.A.T., K.K., K.H.S.M, S.A.G., S.R., H.H., O.T.M., P.S., D.F.G., T.R., G.T., P.M., G.L.N., I.J., and K.S. declare competing interests as employees of deCODE genetics/Amgen. The remaining authors declare no competing interests.

## Additional information

Saedis Saevarsdottir [1,2,3] ✉, Kristbjörg Bjarnadottir[1], Thorsteinn Markusson[1,2], Jonas Berglund[1], Thorunn A. Olafsdottir [1,2], Gisli H. Halldorsson [1,4], Gudrun Rutsdottir[1,4], Kristbjorg Gunnarsdottir[1], Asgeir Orn Arnthorsson[1], Sigrun H. Lund [1], Lilja Stefansdottir[1], Julius Gudmundsson[1], Ari J. Johannesson[3], Arni Sturluson[1], Asmundur Oddsson [1], Bjarni Halldorsson [1], Björn R. Ludviksson[2,5], Egil Ferkingstad [1], Erna V. Ivarsdottir [1,4], Gardar Sveinbjornsson [1], Gerdur Grondal[2,3], Gisli Masson[1], Grimur Hjorleifsson Eldjarn [1], Gudmundur A. Thorisson[1], Katla Kristjansdottir [1], Kirk U. Knowlton[6,7], Kristjan H. S. Moore [1], Sigurjon A. Gudjonsson[1], Solvi Rognvaldsson[1], Stacey Knight [6], Lincoln D. Nadauld[8], Hilma Holm[1], Olafur T. Magnusson[1], Patrick Sulem [1], Daniel F. Gudbjartsson [1,4], Thorunn Rafnar [1], Gudmar Thorleifsson [1], Pall Melsted[1,4], Gudmundur L. Norddahl[1], Ingileif Jonsdottir [1,2,5] & Kari Stefansson [1,2] ✉

[1]deCODE genetics/Amgen, Inc., Reykjavik, Iceland. [2]Faculty of Medicine, School of Health Sciences, University of Iceland, Reykjavik, Iceland. [3]Department of Medicine, Landspitali, the National University Hospital of Iceland, Reykjavik, Iceland. [4]School of Engineering and Natural Sciences, University of Iceland, Reykjavik, Iceland. [5]Department of Immunology, Landspitali, the National University Hospital of Iceland, Reykjavik, Iceland. [6]Intermountain Medical Center, Intermountain Heart Institute, Salt Lake City, UT, USA. [7]School of Medicine, University of Utah, Salt Lake City, UT, USA. [8]Precision Genomics, Intermountain Healthcare, Saint George, UT, USA. ✉e-mail: saedis.saevarsdottir@decode.is; kstefans@decode.is

