## [Peer Review File · Nature Communications]

Start codon variant in LAG3 is associated with decreased LAG-3 expression and increased risk of autoimmune thyroid diseaseREVIEWER COMMENTS

Reviewer #1 (Remarks to the Author):

This analysis of LAG-3 variants and association with autoimmune thyroid disease is noteworthy in the novelty of the genes identified, uses a rigorous method to link the genetic variants with mRNA expression, protein translation, and T-cell and B-cell function, and frames these findings in clinical context, although limited. However, there are five major items that should be addressed before considering for publication and numerous minor edits that would improve the manuscript:

The first major comment is to re-focus the manuscript primarily on the novel LAG-3 start codon variant (rs781745126-T) and build the analysis around this more specifically. Currently, the abstract positions this variant, along with other novel LAG-3 variant and the ZAP-70 variants; however, the analysis only provides substantial data for rs781745126-T in terms of detailed description of the genetics, the mRNA expression, protein expression, T-cell and B-cell function, and subsequent clinical impact. The other LAG-3 and ZAP-70 variants do not have the same degree of supporting information to justify inclusion beyond some preliminary and hypothesis generating findings. These findings for these other variants are indeed interesting, but not at the same level of rigorous analysis as rs781745126-T.

The second major comment is to expand on the clinical impact, if possible, or more clearly state if this is not possible. In particular, a similar analysis of the clinical status of the heterozygotic carriers of rs781745126-T, similar to what was performed for the homozygotic carriers, would provide a more robust analysis of the potential clinical impact of these findings. Also, lines 990-91 could also address a review of LAG-3 variants associated with T1DM and/or vitiligo. Further discussion here would also be helpful regarding the impact of these findings on AITD risk, diagnosis, and management; the association of these findings with previously reported variants associated with thyroid dysfunction associated with checkpoint inhibitor therapy; and ultimately how these findings could impact development and/or use of Anti-LAG-3 therapeutics (these are indeed mentioned, but the degree of discussion and broader literature review could be improved).

The third major comment is to delve further into the founder effects of these variants, describe further the differences between the Icelandic and Finnish variants, and to also address limitations of these founder effects on applicability to the broader population. These founder effects are indeed interesting, but the population is small and there is no mention of limitations or the unique applicability to these populations.

The fourth major comment is that this analysis would benefit from the use of sub-sections and corresponding headers. In the current version, a mix of background info, methods, results, and conclusions are strung together. It would be much easier to follow and digest the info if presented in a clearer flow of background evidence, description of the genetics, the impact on translation and transcription, the impact on T-cell and B-cell function, and then finally the clinical significance and study limitations. And per the first major comment above, a section more focused on the preliminary info of the second LAG-3 and ZAP-70 variants would also be helpful to share these interesting findings but to appropriately separate them from the more robust analysis performed for rs781745126-T. There is no doubt the findings of these additional variants are interesting and potentially clinically impactful, but they should be separated from the main findings and extensive work done on rs781745126-T.

The fifth major comment regards the conclusions. This is actually the most succinct portion of the manuscript and nicely summarizes the findings for the main LAG-3 variant rs781745126-T. However, this is missing any discussion of limitations of the study and how these results could actually impact drug development and/or patient care (the last sentence is indeed true but very generic and not clearly demonstrative of impact). And as noted above, the findings for the second LAG-3 and ZAP-70 variants are very interesting, but not at the same level as for rs781745126-T. The conclusions could be

a good opportunity to identify these limitations too and call for further follow up of these preliminary findings.

Numerous minor comments that would greatly improve readability and understanding of this work:

In the abstract:

The first sentence of abstract is grammatically incorrect and also uses incorrect punctuation. Please fix the sentence fragment (it looks like it should state...290 variants "are" associated with AITD...and also change the semi-colon to a comma.

Please consider moving the statement about the homozygous carriers down to after the two sentences about the novel start codon and the reduced LAG-3 mRNA expression, so it is more of a concluding statement. Also please state in this sentence if these patients have any cancer history, which is included further down in the main section, but would be helpful to include here too.

Appropriately limit the description of the second LAG-3 variant and the ZAP-70 variant to preliminary info based on the genetic findings, predicted structural changes, and potential clinical impact, as the results for these variants are not as robust as those for rs781745126-T.

Also please consider reversing order of the concluding statement here to state that these novel variants associated with T-cell function increase risk of AITD, as well as the B-cell impact.

For the main body of the report:

Please state the exact number of cases and controls rather than stating "almost 111 thousand" and "1.1 million".

The paragraph with lines 68-74 states that several variants were found, but only states the two LAG-3 variants. Please consider adding a line about ZAP-70 here to set up the further details in the subsequent paragraphs (and to follow the presentation order and format of the abstract). This should also be in light of the major comments above regarding appropriate positioning of the second LAG-3 and ZAP-70 variants.

The first part of the paragraph with lines 75-81 provides some background on LAG-3 which is helpful. However, this should also contain background on ZAP-70. This whole paragraph should be moved up before line 68 to provide the background info before going into the findings. Also, there is further introductory info in lines 125-133 which could be moved up here to a more comprehensive intro section rather than jumping around.

Lines 90-91 should state if LAG-3 variants are associated with T1DM or vitiligo. Also, it would be very informative to provide any available medical history on the 416 heterozygotes here and whether any of them had AID and/or cancer, as was stated for the homozygotes. It seems like this might be partially addressed in lines 99-104 in patients with anti-thyroid antibodies, but could be built out further for all of the heterozygotes (or at least stated if info not available).

Also, in line 82-91 it should state where the Icelandic variant occurs structurally as is stated for the Finnish variant, as this is important to further understand the biological plausibility of the findings.

For the figures:

Fig 2 b and c use same representation of plasma levels for homozygotes and heterozygotes. Fine to use both but not one of each. Also clarify if b and c are both only in Icelanders.

Figures 3-5 would also benefit from inclusion of any corresponding info on the homozygotes.

Extended data figure 3 needs correction of "where matched" to "were matched".

Please check use of "motif" versus "motive" throughout.

Reviewer #2 (Remarks to the Author):

The authors previously reported 99 sequence variants associated with a risk of AITD by using a GWAS from Iceland and the UK Biobank. In this GWAS meta-analysis, they found 290 sequence variants by adding GWAS from FinnGen ver 8 (Finland) and Intermountain (USA).

In this study, they found a rare LAG-3 gene variant to be associated with AITD and provided in vitro validation of the function of this gene.

As the authors noted, while LAG-3 is likely to play an important role in the tumor microenvironment as a negative regulator of CD4+ and CD8+ T cells, unlike other molecules of TME cells, no functional association has been identified in the LAG-3 gene to date, and this is important first study to report this.

There are a few minor concerns that needed to be clarified, including the following points.

1. The risk effects of the top variant LAG-3 rs7817451126 in this study is 3.42, while the maximum OR in previous study was only 1.46 for FLT3 rs76428106 (it is 1.34 in this new meta-analysis).

Considering that the Iceland data was also included in the previous study, can you provide any additional explanation as to why this variable was not reported in the previous study and how the concept of the founder effect can be applied? Given the very low MAF of LAG-3 rs7817451126, could the authors elaborate on this in the context of overall AITD risk?

In addition, it would be better to describe a Discussion that compares the risk variants in this paper to those reported in the 2020 paper (especially the top variants in both papers) to help readers understand.

Why did the PTPN22 risk allele change from rs2476601-A to rs2476601-G compared to the previous paper?

2. Considering the mechanistic explanations provided for LAG3 variants, can the authors discuss whether similar mechanisms could be applicable to any CTLA-4 gene SNPs? (although the OR of CTLA-4 is only 1.05 in this study)

3. Could the authors elaborate on the functional role of LAG3 in plasma and on the cell surface, as this would provide a deeper understanding of how changes in LAG3 expression may affect T-cell activation and contribute to AITD risk.

4. The number of subjects in the manuscript can be confusing to readers. In Table 1, the Somascan dataset from Iceland (AITD?) is listed as N=39,155 (MAF 0.13%). However, on line 585, the number of AITD cases in Icelanders is reported as 18,652 (MAF 0.13%). To clarify this, it would be helpful to provide the number of risk variants between AITD and controls (or the MAF of risk variants in AITD and controls separately) in each cohort from Iceland, the UK, Finland, and the US.

Replies to reviewers

Reviewer #1 (Remarks to the Author):

This analysis of LAG-3 variants and association with autoimmune thyroid disease is noteworthy in the novelty of the genes identified, uses a rigorous method to link the genetic variants with mRNA expression, protein translation, and T-cell and B-cell function, and frames these findings in clinical context, although limited. However, there are five major items that should be addressed before considering for publication and numerous minor edits that would improve the manuscript:

1. The first major comment is to re-focus the manuscript primarily on the novel LAG-3 start codon variant (rs781745126-T) and build the analysis around this more specifically. Currently, the abstract positions this variant, along with other novel LAG-3 variant and the ZAP-70 variants; however, the analysis only provides substantial data for rs781745126-T in terms of detailed description of the genetics, the mRNA expression, protein expression, T-cell and B-cell function, and subsequent clinical impact. The other LAG-3 and ZAP-70 variants do not have the same degree of supporting information to justify inclusion beyond some preliminary and hypothesis generating findings. These findings for these other variants are indeed interesting, but not at the same level of rigorous analysis as rs781745126-T.

→We have now re-focused the manuscript and present the findings under subheadings (see details under item 4) so it has more clear focus now on the *LAG3* rs781745126-T variant and the other variants are summarized in a more limited way thereafter, as suggested. We have also replaced the description of these variants in the abstract with more details on the clinical impact of rs781745126-T (see further details in replies to comments 2, 4 and 8 below).

2. The second major comment is to expand on the clinical impact, if possible, or more clearly state if this is not possible. In particular, a similar analysis of the clinical status of the heterozygotic carriers of rs781745126-T, similar to what was performed for the homozygotic carriers, would provide a more robust analysis of the potential clinical impact of these findings. Also, lines 990-91 could also address a review of LAG-3 variants associated with T1DM and/or vitiligo. Further discussion here would also be helpful regarding the impact of these findings on AITD risk, diagnosis, and management; the association of these findings with previously reported variants associated with thyroid dysfunction associated with checkpoint inhibitor therapy; and ultimately how these findings could impact development and/or use of Anti-LAG-3 therapeutics (these are indeed mentioned, but the degree of discussion and broader literature review could be improved).

→ We thank the reviewer for the opportunity to expand this part of the manuscript. The clinical impact of carrying the *LAG3* variants has now been further described, but there were no significant associations with autoimmune diseases or with cancer after multiple testing correction. However, we agree that the observation of concurrent T1DM and vitiligo in one of the three homozygous carriers of rs781745126-T merits special attention, and as shown in Supplementary Table 3, there is a five-fold increased risk of vitiligo among carriers which is nominally significant and survives multiple testing correction for these disease entities ($OR=5.10$, $P=6.5\times 10^{-3}$), while no significant association is observed with type 1 diabetes. It should be noted that the low frequency of rs781745126-T limits the power to detect associations of lower magnitude or in less common diseases. Thus, although for example 2 of three homozygous carriers and 63 out of 416 carriers of rs781745126-T or 15% had a history of cancer diagnosis, this was not significant compared with non-carriers, but we have now added these numbers and described the relevance since LAG-3 inhibitors are an effective cancer therapy. We find no reported associations of LAG3 variants with T1DM or vitiligo, but find one article about accelerated autoimmune diabetes in LAG-3 absence in mice (PMID: 21873518). Since we limit our description to human findings, this reference is not included. The 419 rs781745126-T carriers also had, as expected, several other diagnoses, but as these were not more common among carriers than in the control population, those results were not included. Immune-related adverse events have been associated with longer survival in patients receiving checkpoint inhibitor therapy, so we also checked whether rs781745126-T associated with survival in 9 solid cancer subtypes where information is available, without significant findings, but this is now added as Supplementary Table 4.

The manuscript text has now been expanded as follows:

Abstract: „All three homozygous carriers of rs781745126-T have AITD, of whom one also has two other T-cell mediated diseases, that is vitiligo and type 1 diabetes, and two have a history of cancer. rs781745126-T associates nominally with vitiligo ($OR=5.1$, $P=6.5\times 10^{-3}$) but not with type 1 diabetes. Thus, the effect of rs781745126-T is akin to drugs that inhibit LAG-3, which unleash immune responses and can have thyroid dysfunction and vitiligo as adverse events. This illustrates how a multiomics approach can reveal potential drug targets and safety concerns.“

Results, with new subheading: „Clinical impact of the Icelandic LAG3 rs781745126-T variant.

We identified 416 heterozygous and three homozygous carriers of the rs781745126-T variant in Iceland. The homozygous carriers were all alive, had children and were 65, 70 and 85 years old at the time of analysis. Strikingly, all three homozygotes for the rs781745126-T variant have AITD and one also has two other autoimmune diseases, type 1 diabetes and vitiligo. We therefore tested the association of rs781745126-T with type 1 diabetes and vitiligo, and found a nominally increased risk of vitiligo among rs781745126-T carriers ($OR=5.10$, $P=6.5\times 10^{-3}$) but no association with type 1 diabetes ($OR=1.80$, $P=0.12$). Further testing of 17 other autoimmune diseases yielded no significant association after multiple testing correction (Supplementary Table 3). Two of the three homozygous carriers of rs781745126-T and 15% of the heterozygous carriers have a history of cancer, but there was no association of rs781745126-T with cancer overall ($N=49,981$; $P=0.27$) or any of 31

cancer subtypes or with cancer survival, retrieved from nationwide data (Supplementary Table 4, $P > 0.05$ for all).“

Discussion:

„There are currently several drugs inhibiting LAG-3 registered or in clinical trials to treat cancer{Sauer, 2022}. However, unlike its precursor immune checkpoint inhibitor drugs targeting CTLA-4, PD-1 and PD-L1{Aggarwal, 2023}, LAG-3 drug development has hitherto not had any genetic support for disease association. CTLA-4 agonist (abatacept) is used to treat autoimmune diseases{Maruhashi, 2020} and common sequence variants at the loci of the CTLA4 and CD274 genes, encoding the CTLA-4 and PD-L1 checkpoint proteins, are previously reported to associate with AITD and related phenotypes, and confirmed in our meta-analysis (rs911760 and rs11571297, see Supplementary Table 1). Although there are two independent signals at the CTLA4 locus in our current meta-analysis, they did not fulfill the criteria in the multiomics analysis for CTLA4 being a candidate causal gene. Nevertheless, the whole genome sequencing data available in our study populations (Methods) shows that the lead signals associate with structural variants in CTLA4, thereby pointing to CTLA4 as a candidate gene in AITD, where rs11571297 represents a common 42 base-pair deletion in the 3' untranslated region (UTR) of CTLA4, while there are several less frequent deletions of variable length. This is in line with a previous report of variations in the length of dinucleotide (AT) repeats within the 3'UTR of CTLA4, where repeat length correlated with mRNA and protein levels of CTLA-4{de Jong, 2016}. Our observation that all three homozygotes for the LAG3 rs781745126-T variant have AITD indicates a high penetrance of the variant. Interestingly, one of the homozygotes also has two other T-cell mediated autoimmune diseases, type 1 diabetes and vitiligo, that like thyroid dysfunction, are known immune-related side effects of anti-LAG-3 and the other checkpoint inhibitors{Yoo, 2023;Ding, 2022;Ghani, 2023; Aggarwal, 2023}. rs781745126-T confers a fivefold increased risk of vitiligo, but neither of the LAG3 variants associated with other autoimmune diseases or with cancer after multiple testing correction, although the power to detect associations of lower magnitude or in less common diseases is limited for such rare variants.

A substantial proportion of patients receiving immune checkpoint inhibitor therapy experience thyroid dysfunction, but it is interesting to note that immune related adverse events have been associated with longer survival in treated patients{Spiers, 2019;Miao, 2024}. Furthermore, patients receiving PD-1 or PD-L1 inhibitors that have high polygenic risk score for hypothyroidism are reported to have longer survival but also a higher risk of hypo- and hyperthyroidism as well as vitiligo{Khan, 2021}. In the Icelandic study population, information about survival is available for 12 cancer subtypes, but rs781745126-T did not associate with survival after multiple testing correction.

Although it may not be practical to test rare variants as clinical predictors of immune-related adverse events in patients receiving anti-LAG3 inhibitors, our findings provide important evidence for the potential adverse events on LAG-3 inhibiting therapy, since the effect of rs781745126-T among carriers is similar as the effect of LAG-3 inhibiting therapy, and illustrates the power of a multiomics approach to reveal not only potential drug targets but also safety concerns.“

3. The third major comment is to delve further into the founder effects of these variants, describe further the differences between the Icelandic and Finnish variants, and to also address limitations of these founder effects on applicability to the broader population. These founder effects are indeed interesting, but the population is small and there is no mention of limitations or the unique applicability to these populations.

→We have now added in both the results and the new discussion chapter, including the conclusion, further information about this important topic, as follows:

Results: „We identified several sequence variants with a strong effect on AITD risk (Fig. 1b). Two of these are in the same gene, LAG3; a rare 5'-UTR variant (rs781745126-T, OR=3.42, $P=2.2\times 10^{-16}$) and a missense variant (rs149722682-A, Pro67Thr, OR=2.11, $P=2.2\times 10^{-9}$). These two sequence variants in LAG3 illustrate a founder effect in Iceland (rs781745126-T, minor allele frequency (MAF)=0.13% in Iceland, 0.00016% in UK and no carriers found in other study populations) and in Finland (rs149722682-A, MAF=0.10%, no carriers found outside Finland). A missense variant in the ZAP70 gene (rs145955907-T, Thr155Met) that associates with an increased risk of AITD (OR=1.27, $P=1.0\times 10^{-16}$) is also an example of a founder effect in Finland (MAF=1.95%), although it is present, but rare, in all the other populations (MAF=0.01-0.03%, see Supplementary Information 8).“

Discussion: „The rs781745126-T and Pro67Thr variants in LAG3 that associate with an increased risk of AITD in Iceland and Finland respectively, exemplify a founder effect in these populations, that have been rather isolated for centuries. Rare variants may not explain a large proportion of AITD cases in the overall population, but the risk is substantial among carriers and these variants point to LAG3 as a candidate gene through reduced transcription and protein levels on one hand, and altered protein structure on the other hand, indicating that LAG-3 reduction predisposes to AITD. This demonstrates the power of bottlenecked populations to identify rare disease-associated variants with high risk, that provide insight into the pathogenesis and potential identification of drug targets {Kurki, 2023;Helgason, 2003}.

Another example of population enrichment of rare variants is ZAP70 Thr155Met, which has a minor allele frequency of 1.95% in Finland and only 0.01-0.03% in Iceland, UK and USA, resulting in higher statistical power to detect an association with AITD in Finland than in the other populations.“

4. The fourth major comment is that this analysis would benefit from the use of subsections and corresponding headers. In the current version, a mix of background info, methods, results, and conclusions are strung together. It would be much easier to follow and digest the info if presented in a clearer flow of background evidence, description of the genetics, the impact on translation and transcription, the impact on T-cell and B-cell function, and then finally the clinical significance and study limitations. And per the first major comment above, a section more focused on the preliminary info of the second LAG-3 and ZAP-70 variants would also be helpful to share these interesting findings but to appropriately separate them from the more robust analysis performed for rs781745126-T. There is no doubt the findings of these additional variants are

interesting and potentially clinically impactful, but they should be separated from the main findings and extensive work done on rs781745126-T.

→We thank the reviewer for this suggestion and have added subheadings, and highlighted with a more clear focus the *LAG3* rs781745126-T start codon variant in the manuscript, while the other variants are summarized with more limited details thereafter, as suggested. In addition, we have summarized the order of analyses in the end of the background text and separated and expanded the discussion chapter to address several of the other comments.

5. The fifth major comment regards the conclusions. This is actually the most succinct portion of the manuscript and nicely summarizes the findings for the main LAG-3 variant rs781745126-T. However, this is missing any discussion of limitations of the study and how these results could actually impact drug development and/or patient care (the last sentence is indeed true but very generic and not clearly demonstrative of impact). And as noted above, the findings for the second LAG-3 and ZAP-70 variants are very interesting, but not at the same level as for rs781745126-T. The conclusions could be a good opportunity to identify these limitations too and call for further follow up of these preliminary findings.

→In order to address this important point, we have now re-formatted the manuscript so there is a separate discussion chapter where we address these suggestions, as well as other comments that are raised. Although too long to include here in the replies all changes, the following paragraphs address the points that are highlighted in this comment, as follows:

„There are currently several drugs inhibiting LAG-3 registered or in clinical trials to treat cancer{Sauer, 2022}. However, unlike its precursor immune checkpoint inhibitor drugs targeting CTLA-4, PD-1 and PD-L1{Aggarwal, 2023}, LAG-3 drug development has hitherto not had any genetic support for disease association. CTLA-4 agonist (abatacept) is used to treat autoimmune diseases{Maruhashi, 2020} and common sequence variants at the loci of the CTLA4 and CD274 genes, encoding the CTLA-4 and PD-L1 checkpoint proteins, are previously reported to associate with AITD and related phenotypes, and confirmed in our meta-analysis (rs911760 and rs11571297, see Supplementary Table 1). Although there are two independent signals at the CTLA4 locus in our current meta-analysis, they did not fulfill the criteria in the multiomics analysis for CTLA4 being a candidate causal gene. Nevertheless, the whole genome sequencing data available in our study populations (Methods) shows that the lead signals associate with structural variants in CTLA4, thereby pointing to CTLA4 as a candidate gene in AITD, where rs11571297 represents a common 42 base-pair deletion in the 3' untranslated region (UTR) of CTLA4, while there are several less frequent deletions of variable length. This is in line with a previous report of variations in the length of dinucleotide (AT) repeats within the 3'UTR of CTLA4, where repeat length correlated with mRNA and protein levels of CTLA-4{de Jong, 2016}.

Our observation that all three homozygotes for the LAG3 rs781745126-T variant have AITD indicates a high penetrance of the variant. Interestingly, one of the homozygotes also has two other T-cell mediated autoimmune diseases, type 1 diabetes and vitiligo, that like thyroid dysfunction, are known immune-related side effects of anti-LAG-3 and the other checkpoint inhibitors{Yoo, 2023;Ding, 2022;Ghani, 2023; Aggarwal, 2023}. rs781745126-T confers a

fivefold increased risk of vitiligo, but neither of the LAG3 variants associated with other autoimmune diseases or with cancer after multiple testing correction, although the power to detect associations of lower magnitude or in less common diseases is limited for such rare variants.

A substantial proportion of patients receiving immune checkpoint inhibitor therapy experience thyroid dysfunction, but it is interesting to note that immune related adverse events have been associated with longer survival in treated patients{Spiers, 2019;Miao, 2024}. Furthermore, patients receiving PD-1 or PD-L1 inhibitors that have high polygenic risk score for hypothyroidism are reported to have longer survival but also a higher risk of hypo- and hyperthyroidism as well as vitiligo{Khan, 2021}. In the Icelandic study population, information about survival is available for 9 cancer subtypes, but rs781745126-T did not associate with survival after multiple testing correction (Supplementary Table 4).

Although it may not be practical to test rare variants as clinical predictors of immune-related adverse events in patients receiving anti-LAG3 inhibitors, our findings provide important evidence for the potential adverse events on LAG-3 inhibiting therapy, since the effect of rs781745126-T among carriers is similar as the effect of LAG-3 inhibiting therapy, and illustrates the power of a multiomics approach to reveal not only potential drug targets but also safety concerns.

.....

Compared to our previously reported meta-analysis of 30 thousand AITD cases and 725 thousand controls in study populations from Iceland and UK{Saevarsdottir, 2020}, diagnoses registered in primary care and by private practitioners were now available in Iceland (as well as in UK, the USA and Finland). This resulted in almost four times larger study population of AITD, both overall and the Icelandic subset.

.....

We did not have power to detect the association of the Icelandic rs7817451126-T variant with AITD in our previous report, while it reached the genome-wide significance level in the current study, highlighting the importance of large study populations to detect associations with rare variants, especially those that are enriched in certain populations. The rs781745126-T and Pro67Thr variants in LAG3 that associate with an increased risk of AITD in Iceland and Finland respectively, exemplify a founder effect in these populations, that have been rather isolated for centuries. Rare variants may not explain a large proportion of AITD cases in the overall population, but the risk is substantial among carriers and these variants point to LAG3 as a candidate gene through reduced transcription and protein levels on one hand, and altered protein structure on the other hand, indicating that LAG-3 reduction predisposes to AITD. This demonstrates the power of bottlenecked populations to identify rare disease-associated variants with high risk, that provide insight into the pathogenesis and potential identification of drug targets{Kurki, 2023;Helgason, 2003}.

Another example of population enrichment of rare variants is ZAP70 Thr155Met, which has a minor allele frequency of 1.95% in Finland and only 0.01-0.03% in Iceland, UK and USA, resulting in higher statistical power to detect an association with AITD in Finland than in the other populations.

.....

Other variants in the LAG3, ZAP70 and other genes, which result in amino-acid changes at conserved protein regions and are likely alter the conformation, call for further investigation.“

Numerous minor comments that would greatly improve readability and understanding of this work:

In the abstract:

6. The first sentence of abstract is grammatically incorrect and also uses incorrect punctuation. Please fix the sentence fragment (it looks like it should state...290 variants “are” associated with AITD...and also change the semi-colon to a comma.

→ Thank you, we have now corrected this.

7. Please consider moving the statement about the homozygous carriers down to after the two sentences about the novel start codon and the reduced LAG-3 mRNA expression, so it is more of a concluding statement. Also please state in this sentence if these patients have any cancer history, which is included further down in the main section, but would be helpful to include here too.

→ We have now moved this sentence as suggested, and added the following text (also a reply to comment 2 above):

„All three homozygous carriers of rs781745126-T have AITD, of whom one also has two other T-cell mediated diseases, that is vitiligo and type 1 diabetes. rs781745126-T associates nominally with vitiligo (OR=5.1, P=6.5×10⁻³) but not with type 1 diabetes.“

8. Appropriately limit the description of the second LAG-3 variant and the ZAP-70 variant to preliminary info based on the genetic findings, predicted structural changes, and potential clinical impact, as the results for these variants are not as robust as those for rs781745126-T.

→ In light of this and other comments, we have now replaced the description of these variants in the abstract, with a more detailed description of the potential clinical impact of the findings, in line with the manuscript conclusions (as mentioned under comment 5 above), see additions in reply to comment 7 and as follows:

„Thus, the effect of rs781745126-T is akin to drugs that inhibit LAG-3, which unleash immune responses and can have thyroid dysfunction and vitiligo as adverse events. This illustrates how a multiomics approach can reveal potential drug targets and safety concerns.“

9. Also please consider reversing order of the concluding statement here to state that these novel variants associated with T-cell function increase risk of AITD, as well at the B-cell impact.

→ See suggested changes above in the reply to comment 8, and we have also expanded this statement earlier in the abstract, as follows: *„Multiomics analysis yielded 235 candidate genes outside the MHC-region and the findings highlight the importance of genes involved in T-cell regulation.“* Regarding the B-cell impact, we did see an effect of the LAG3 variant on immortalized B-cells, but have limited space in the abstract to describe that further, and in the sentence above, we are only referring to the candidate genes as a whole, as we modified the abstract in line with comments including comment 5 which refers to the manuscript conclusion as a nice summary, wherefrom this sentence is retrieved. Then, we discuss this in more detail in the manuscript text, and have included the B-cell impact in the conclusion of the discussion chapter, as follows: *„The Icelandic LAG3 variant generates a novel start codon for protein coding and results in a reduced capacity to induce mRNA expression of LAG3 in T-cell subsets. Both activated and exhausted T-cells as well as immortalized B-cells have lower expression of LAG-3 on their surface and in cell supernatant.“*

For the main body of the report:

10. Please state the exact number of cases and controls rather than stating “almost 111 thousand” and “1.1 million”.

→ We have changed this to the exact numbers of 110,945 cases and 1,084,290 controls.

11. The paragraph with lines 68-74 states that several variants were found, but only states the two LAG-3 variants. Please consider adding a line about ZAP-70 here to set up the further details in the subsequent paragraphs (and to follow the presentation order and format of the abstract). This should also be in light of the major comments above regarding appropriate positioning of the second LAG-3 and ZAP-70 variants.

→ We have now added the following sentence, as suggested: *„A missense variant in the ZAP70 gene (rs145955907-T, Thr155Met) that associates with an increased risk of AITD (OR=1.27, P=1.0×10⁻¹⁶) is also an example of a founder effect in Finland (MAF=1.95%), although it is present, but rare, in all the other populations (MAF=0.01-0.03%, see Supplementary Information 8).“*

12. The first part of the paragraph with lines 75-81 provides some background on LAG-3 which is helpful. However, this should also contain background on ZAP-70. This whole paragraph should be moved up before line 68 to provide the background info before going into the findings. Also, there is further introductory info in lines 125-133 which could be moved up here to a more comprehensive intro section rather than jumping around.

→ We have now moved the text about ZAP-70 as suggested after the text about LAG-3. However, since this is a hypothesis-free study, we do not find it appropriate to move the text about the protein products of the genes in which the sequence variants associated with AITD are, before we present which sequence variants are found to associate with AITD, but we agree that a more structured presentation is preferable. Therefore, we have, also in line with

other comments, divided the text into sub-sections and corresponding headers, and provided a separate discussion chapter where we find this text to be most suitably placed, with further discussion of the findings in the context of previous evidence.

13. Lines 90-91 should state if LAG-3 variants are associated with T1DM or vitiligo. Also, it would be very informative to provide any available medical history on the 416 heterozygotes here and whether any of them had AID and/or cancer, as was stated for the homozygotes. It seems like this might be partially addressed in lines 99-104 in patients with anti-thyroid antibodies, but could be built out further for all of the heterozygotes (or at least stated if info not available).

→ The clinical impact of carrying the *LAG3* variants has now been further described (see also comments 2 and 5 above) and we observe a fivefold increased risk of vitiligo among rs781745126-T carriers, which is nominally significant despite the few cases (n=428) of this rare disease in Iceland. There were no other significant associations with autoimmune diseases or with cancer observed for the *LAG3* variants after multiple testing correction. It should be noted that their low frequency limits the power to detect associations of lower magnitude or in less common diseases. Thus, although for example 4 out of 416 heterozygous carriers have type 1 diabetes, 3 have psoriasis and a few others have at least one autoimmune disease, none of these findings are significant and we summarize the findings in Supplementary Table 3. For cancer, 63 out of 416 heterozygous carriers of rs781745126-T or 15% had a history of cancer diagnosis, and 2 of three homozygous carriers, but the association was not significant compared with non-carriers, but we have now added the numbers to the text and summarized the findings in Supplementary Tables 3-4. We have information about all other diagnoses of the AITD cases in Iceland, but none of these have significant association with rs781745126-T, and we thought it was preferable to limit the analyses with autoimmune disease and cancer, since an addition of other diseases in a phenome-wide study calls for a stricter multiple testing correction threshold of significance, but we can add such data upon request by the Editor. Information about thyroid autoantibody measurement was available in a nationwide Icelandic database since 2005 (8,196 individuals measured of whom 4,667 have AITD and 2,442 had thyroid autoantibodies at that timepoint), illustrating that it is not routine praxis to measure thyroid autoantibodies (and not required in clinical guidelines) and the tests of those often become negative over time. Therefore, it is preferable to analyse the medical history in all AITD cases and not in the subset with available autoantibody measurement, and we have now clarified this in the manuscript and below Supplementary Table 3 and we have also added association analyses of the Finnish *LAG3* rs149722682 (Pro67Thr) variant with autoimmune diseases and cancer to that table, and added Supplementary Table 4 with results for cancer subtypes and survival, with available data in Iceland.

The manuscript text has been modified as follows:

Results: „We therefore tested the association of rs781745126-T with type 1 diabetes and vitiligo and found a nominally increased risk of vitiligo among rs781745126-T carriers (OR=5.10, P=6.5×10⁻³) but no association with type 1 diabetes (OR=1.80, P=0.12). Further testing of 17 other autoimmune diseases yielded no significant association after multiple testing correction (Supplementary Table 3). Two of the three homozygous carriers of rs781745126-T and 15% of the heterozygous carriers had a history of cancer, but there was

*no association of rs781745126-T with cancer overall (N=49,981, P=0.27) or any of 31 cancer subtypes retrieved from nationwide data (Supplementary Table 4, P>0.05 for all). In an Icelandic cohort with thyroid autoantibody measurements (8,196 individuals measured of whom 4,667 have AITD and 2,442 had thyroid autoantibodies at that timepoint), we found that among those who had thyroid autoantibodies (N=1967, Supplementary Table 3), the risk of AITD was 63-fold higher in rs781745126-T carriers than in non-carriers (P=8.2×10⁻³).“
„The rare Finnish Pro67Thr did not associate with other autoimmune diseases or with cancer, after multiple testing correction (Supplementary Table 3).“*

Discussion: „Our observation that all three homozygotes for the LAG3 rs781745126-T variant have AITD indicates a high penetrance of the variant. Interestingly, one of the homozygotes also has two other T-cell mediated autoimmune diseases, type 1 diabetes and vitiligo, that like thyroid dysfunction, are known immune-related side effects of anti-LAG-3 and the other checkpoint inhibitors{Yoo, 2023;Ding, 2022;Ghani, 2023; Aggarwal, 2023}. rs781745126-T confers a fivefold increased risk of vitiligo, but neither of the LAG3 variants associated with other autoimmune diseases or with cancer after multiple testing correction, although the power to detect associations of lower magnitude or in less common diseases is limited for such rare variants.

A substantial proportion of patients receiving immune checkpoint inhibitor therapy experience thyroid dysfunction, but it is interesting to note that immune related adverse events have been associated with longer survival in treated patients{Spiers, 2019;Miao, 2024}. Furthermore, patients receiving PD-1 or PD-L1 inhibitors that have high polygenic risk score for hypothyroidism are reported to have longer survival but also a higher risk of hypo- and hyperthyroidism as well as vitiligo{Khan, 2021}. In the Icelandic study population, information about survival is available for 9 cancer subtypes, but rs781745126-T did not associate with survival after multiple testing correction (Supplementary Table 4).

Although it may not be practical to test rare variants as clinical predictors of immune-related adverse events in patients receiving anti-LAG3 inhibitors, our findings provide important evidence for the potential adverse events on LAG-3 inhibiting therapy, since the effect of rs781745126-T among carriers is similar as the effect of LAG-3 inhibiting therapy, and illustrates the power of a multiomics approach to reveal not only potential drug targets but also safety concerns.“

14. Also, in line 82-91 it should state where the Icelandic variant occurs structurally as is stated for the Finnish variant, as this is important to further understand the biological plausibility of the findings.

→We have now explained this further, but felt it was more logical to place it under the new subheading: „Impact of LAG3 rs781745126-T on transcription, translation and protein levels“, with the following additions to the text: „As illustrated in Fig. 2a, rs781745126-T is in the 5' untranslated region (UTR) of the LAG3 gene, which is a regulatory region that is transcribed into mRNA but not translated into protein. We found that rs781745126-T generates a novel translation initiation site (TIS) 303 bp upstream of the canonical TIS. The new TIS or start codon has a stronger Kozak sequence, that is a stronger nucleic acid motif for the initiation of protein translation{Leppek, 2018}. It is predicted to generate a novel upstream open reading frame (uORF) that contains a STOP codon after 84 codons or 51bp

upstream of the canonical LAG-3 protein TIS. Such uORFs have been shown to reduce translation of the downstream ORF and, to a lesser degree, destabilize the mRNA (Jia, 2020; Johnstone, 2016). rs781745126-T may therefore reduce the use of the canonical start site, which could result in lower levels of LAG-3.

Discussion: „The rs781745126-T and Pro67Thr variants in LAG3 that associate with an increased risk of AITD in Iceland and Finland respectively, exemplify a founder effect in these populations, that have been rather isolated for centuries. Rare variants may not explain a large proportion of AITD cases in the overall population, but the risk is substantial among carriers and these variants point to LAG3 as a candidate gene through reduced transcription and protein levels on one hand, and altered protein structure on the other hand, indicating that LAG-3 reduction predisposes to AITD.“

For the figures:

15. Fig 2 b and c use same representation of plasma levels for homozygotes and heterozygotes. Fine to use both but not one of each. Also clarify if b and c are both only in Icelanders.

→ The LAG-3 levels were measured in Icelanders, and that has been clarified in the figure legend now. The reason for the different units for LAG3 levels in plasma in Fig. 2b and c is that these are two different methods, and unfortunately it is not possible to convert between these two and present same units for both, but the correlation between the levels measured with two methods is excellent ($R=0.93$, $P<2.2\times 10^{-16}$) and that applies both to the carriers ($R=0.91$, $P<2.2\times 10^{-16}$) and non-carriers ($R=0.84$, $P<2.2\times 10^{-16}$). We think it was important to validate the finding from the aptamer-based SomaScan platform, that LAG-3 levels differed between rs781745126-T carriers and non-carriers, with another method, which was the antibody-based MSD R-PLEX method. On SomaScan, the levels are relative units based on light intensity, that are presented by standardized adjusted beta levels, while pg/ml are measured on the antibody-based MSD R-PLEX method.

16. Figures 3-5 would also benefit from inclusion of any corresponding info on the homozygotes.

→ We agree that it would have been very valuable to measure homozygote individuals as well, but unfortunately we did not have PBMC samples from any of the 3 homozygotes.

17. Extended data figure 3 needs correction of “where matched” to “were matched”.

→ Thank you for noticing this spelling error, it has now been corrected.

18. Please check use of "motif" versus "motive" throughout.

→ We have now corrected this and replaced motive with motif.

Reviewer #2 (Remarks to the Author):

The authors previously reported 99 sequence variants associated with a risk of AITD by using a GWAS from Iceland and the UK Biobank. In this GWAS meta-analysis, they found 290 sequence variants by adding GWAS from FinnGen ver 8 (Finland) and Intermountain (USA).

In this study, they found a rare LAG-3 gene variant to be associated with AITD and provided in vitro validation of the function of this gene.

As the authors noted, while LAG-3 is likely to play an important role in the tumor microenvironment as a negative regulator of CD4+ and CD8+ T cells, unlike other molecules of TME cells, no functional association has been identified in the LAG-3 gene to date, and this is important first study to report this.

There are a few minor concerns that needed to be clarified, including the following points.

1a. The risk effects of the top variant LAG-3 rs7817451126 in this study is 3.42, while the maximum OR in previous study was only 1.46 for FLT3 rs76428106 (it is 1.34 in this new meta-analysis). Considering that the Iceland data was also included in the previous study, can you provide any additional explanation as to why this variable was not reported in the previous study and how the concept of the founder effect can be applied? Given the very low MAF of LAG-3 rs7817451126, could the authors elaborate on this in the context of overall AITD risk? In addition, it would be better to describe a Discussion that compares the risk variants in this paper to those reported in the 2020 paper (especially the top variants in both papers) to help readers understand.

→ The number of cases from Iceland was 4,692 cases in the previous study but increased to 18,651 cases in the current study, after addition of AITD diagnoses registered in primary care and by private practitioners in Iceland (as well as in UK, the USA and Finland). Thus, the study cohort was almost four times larger now and association of the rs7817451126 variant was genome-wide significant and reported now, but did not reach genome-wide significance level in the previous study. Also, as you mention, the association of *FLT3* rs76428106 in the previous meta-analysis, that included a total of 30,234 cases and 725,172 controls from Iceland and UK, was 1.46 with $P=2.37 \times 10^{-24}$ while in the current meta-analysis of larger study populations from these countries as well as study populations from Finland and USA (this low-frequency variant is present in all those study populations), with a total of 110,945 cases and 1,084,290 controls, there is a substantially more significant P -value of 2.30×10^{-44} , and a strong association in all the included study populations, although the OR estimates are somewhat lower in Finland (1.30) and USA (1.16) than in our previous report, resulting in an overall OR of 1.34, compared to 1.46 in our previous report.

The founder effect concept can be applied when rare sequence variants, that are enriched in relatively homogeneous or isolated populations, like in Iceland and Finland, contribute to the risk of common disorders in those populations, and the two sequence variants in the *LAG3* gene that are associated with AITD are examples of that. Although a rare variant like

rs7817451126-T may not explain a large proportion of AITD cases in the overall population, the risk is 3.4-fold among carriers, and all three homozygotes have AITD. Furthermore, the LAG-3 levels of carriers in plasma are low, indicating that when this inhibitory immune check-point protein is lacking, it predisposes to AITD. There is no reason to believe that the relevance in terms of disease risk of carriers is different in other populations, as exemplified by the ZAP70 variant, and therefore the finding is as valuable to understand disease pathogenesis, although such variants explain fewer cases overall in populations where the allele frequency is lower.

We have now added clarifications in the manuscript to explain this as well as the concept of the founder effect, the context of AITD risk and comparison of the top variants in our previous and current reports, as follows:

- Introduction (underlined text was previously only in the methods section): *„Here, we report a meta-analysis of 110,945 cases and 1,084,290 controls, using the same case definition{Saevardsdottir, 2020 #190;Burch, 2019 #59}. In addition to our previous meta-analysis, AITD diagnoses registered in primary care and by private practitioners in Iceland and UK are now included, as well as study populations from the USA and Finland.“*

Discussion: „Compared to our previously reported meta-analysis of 30 thousand AITD cases and 725 thousand controls in study populations from Iceland and UK{Saevardsdottir, Nature 2020}, diagnoses registered in primary care and by private practitioners were now available in Iceland (as well as in UK, the USA and Finland). This resulted in almost four times larger study population, both overall and the Icelandic subset. A low frequency stop mutation in FLT3 (rs76428106-C) had the strongest effect on AITD risk in our previous report{Saevardsdottir, Nature 2020} and we validate this association here (OR=1.34, $P=2.30 \times 10^{-44}$). We did not have power to detect the association of the Icelandic rs7817451126-T variant with AITD in our previous report, while it reached the genome-wide significance level in the current study, highlighting the importance of large study populations to detect associations with rare variants, especially those that are enriched in certain populations. The rs781745126-T and Pro67Thr variants in LAG3 that associate with an increased risk of AITD in Iceland and Finland respectively, exemplify a founder effect in these populations, that have been rather isolated for centuries. Rare variants may not explain a large proportion of AITD cases in the overall population, but the risk is substantial among carriers and these variants point to LAG3 as a candidate gene through reduced transcription and protein levels on one hand, and altered protein structure on the other hand, indicating that LAG-3 reduction predisposes to AITD. This demonstrates the power of bottlenecked populations to identify rare disease-associated variants with high risk, that provide insight into the pathogenesis and potential identification of drug targets{Kurki, 2023;Helgason, 2003}.

Another example of population enrichment of rare variants is ZAP70 Thr155Met, which has a minor allele frequency of 1.95% in Finland and only 0.01-0.03% in Iceland, UK and USA, resulting in higher statistical power to detect an association with AITD in Finland than in the other populations.“

1b) Why did the PTPN22 risk allele change from rs2476601-A to rs2476601-G compared to the previous paper?

→ The *PTPN22* risk allele is still rs2476601-A, with an OR of 1.36, see the GWAS meta-analysis results in Supplementary Table 1 (A=EA=effect allele). However, we thank the reviewer for the careful review, because we realise that in Supplementary Table 2, where sequence variants that point to candidate genes through an effect on protein coding are listed, the effect allele in the output was listed as G, and we have now changed that to A. The same error was in Figure 1 (should be rs2476601-A and not -G) and is now corrected. We apologise for the confusion this may have caused. This does not affect the manuscript text or any findings that are described in the paper.

2. Considering the mechanistic explanations provided for LAG3 variants, can the authors discuss whether similar mechanisms could be applicable to any CTLA-4 gene SNPs? (although the OR of CTLA-4 is only 1.05 in this study)

→ We thank the reviewer for this question, that gave us the opportunity to explain further these signals. For the two lead signals at the locus of the *CTLA4* gene in our GWAS meta-analysis, we do not detect any effect on protein coding, gene expression or protein levels, that point to a gene. Although we know that there are variants (SNPs) at this locus that do have such effect, these variants did not after conditional analysis have independent association with AITD, and could merely be the shadow of the stronger signal, and were therefore not mentioned in the manuscript. Thus, the *CTLA4* missense variant rs231775, that was an independent signal at the locus in our previous paper, has an unadjusted OR of 1.13 and unadjusted *P*-value of 1.04×10^{-129} , but when it is adjusted for one of the lead signals, rs231726, the *P*-value drops to 10^{-11} so it cannot be regarded as a separate independent signal. Furthermore, although rs231775 is correlated with rs231726, the correlation coefficient is $r^2=0.77$ in our study population, which does not fulfil the criteria of $r^2>0.8$ to be regarded as a part of the same signal and be listed in Supplementary Table 2 over coding variants. If it had reached 0.8, this signal would have pointed to *CTLA4* as a candidate gene and be included in Fig. 1. Thus, our data do not support that sequence variants at the *CTLA4* locus that are associated with AITD point to that gene, and we would need a series of functional studies performed on activated PBMCs etc. to understand the mechanistic consequences of the lead signal at the locus, which is in the 3- UTR region, as we did for the lead *LAG3* variant that is in the 5- UTR region.

We have now looked at the whole genome sequencing data available in our study populations (Methods) for the lead signals at the *CTLA4* locus and it turns out that the lead signals associate with multiallelic structural variants, that do thereby point to this as a candidate gene. Exploration of that in order to understand the functional consequences is certainly an interesting topic for further investigation, but not possible to set up within the time-frame given for the revision of this manuscript, but we add a reference to a previous study on that topic.

We have now added an explanation of this in the Discussion chapter: „*There are currently several drugs inhibiting LAG-3 registered or in clinical trials to treat cancer*{Sauer, 2022}. *However, unlike its precursor immune checkpoint inhibitor drugs targeting CTLA-4, PD-1 and PD-L1*{Aggarwal, 2023}, *LAG-3 drug development has hitherto not had any genetic support for disease association. CTLA-4 agonist (abatacept) is used to treat autoimmune diseases*{Maruhasni, 2020} *and common sequence variants at the loci of the CTLA4 and CD274 genes, encoding the CTLA-4 and PD-L1 checkpoint proteins, are previously reported*

to associate with AITD and related phenotypes, and confirmed in our meta-analysis (rs911760 and rs11571297, see Supplementary Table 1). Although there are two independent signals at the CTLA4 locus in our current meta-analysis, they did not fulfill the criteria in the multiomics analysis for CTLA4 as a candidate causal gene. Nevertheless, the whole genome sequencing data available in our study populations (Methods) shows that the lead signals associate with structural variants in CTLA4, thereby pointing to CTLA4 as a candidate gene in AITD, where rs11571297 represents a common 42 base-pair deletion in the 3' untranslated region (UTR) of CTLA4, while there are several less frequent deletions of variable length. This is in line with a previous report of variations in the length of dinucleotide (AT) repeats within the 3'UTR of CTLA4, where repeat length correlated with mRNA and protein levels of CTLA-4{de Jong, 2016}.

3. Could the authors elaborate on the functional role of LAG3 in plasma and on the cell surface, as this would provide a deeper understanding of how changes in LAG3 expression may affect T-cell activation and contribute to AITD risk.

→ We feel that we have already addressed the functional role of the membrane-bound LAG-3, and that description is now in the first and second paragraph of the discussion chapter of the revised manuscript. We have added a description about the soluble form, but much less has been reported on that.

„LAG3 encodes lymphocyte activation gene 3 (LAG-3) protein and rs781745126-T generates a novel start codon for an open reading frame upstream (uORF) to the canonical protein coding sequence of LAG3. Such uORFs have been shown to reduce translation of the downstream ORF{Jia, 2020;Johnstone, 2016}, and may therefore result in lower protein levels. Indeed, we observe that the rs781745126-T variant halves LAG-3 levels in plasma among heterozygotes.

LAG-3 is a co-receptor that is reported to be expressed by activated (CD4⁺, CD8⁺) and regulatory T-cells in humans{Hivroz, 2022}. It is structurally homologous to CD4 and binds to MHC-class II molecules with higher affinity than CD4, thereby preventing antigen mediated T-cell receptor signaling{Maruhasni, 2020}. Recent findings also suggest that LAG-3 could inhibit T-cell receptor signaling in the absence of MHC-class II, which could at least partly explain the inhibitory role LAG-3 has on MHC-class I restricted CD8⁺ T-cells{Hivroz, 2022;Guy, 2022}. Prolonged exposure to antigen and inflammatory signals increases LAG-3 expression and leads to a state of exhaustion in T-cells, resulting in a reduced capacity to kill viruses and malignant cells, while inhibition of LAG-3 enhances viral control and anti-tumor immune responses{Hu, 2020;Shan, 2020}. A proteolytic cleavage of surface LAG-3 creates soluble LAG-3 in serum and plasma, but the functional role of this soluble form is incompletely understood{Li, 2023;Graydon, 2021}. Our results are consistent with that reduced LAG-3 mediated T-cell regulation predisposes to AITD.

4. The number of subjects in the manuscript can be confusing to readers. In Table 1, the Somasca dataset from Iceland (AITD?) is listed as N=39,155 (MAF 0.13%). However, on line 585, the number of AITD cases in Icelanders is reported as 18,652 (MAF 0.13%). To clarify this, it would be helpful to provide the number of risk variants between AITD

and controls (or the MAF of risk variants in AITD and controls separately) in each cohort from Iceland, the UK, Finland, and the US.

→The number of AITD cases in Iceland is, as you state, 18,652, while the Somasca dataset is an Icelandic cohort of cases and controls (irrespective of diseases) that had available plasma samples (N=39,155) and of those, 37,943 also had genotypes and overlaps with the Icelandic cohort of AITD cases and controls (see further description in ref. Ferkingstad, E. et al. Large-scale integration of the plasma proteome with genetics and disease. Nat Genet 2021). We have now added this explanation to the table, as follows: „*rs781745126 is only present in Iceland and rs149722682 in Finland.*“ Furthermore, we have added the results for all risk variants in AITD for each cohort separately with MAF of risk variants to Supplementary Information 8.

REVIEWERS' COMMENTS

Reviewer #1 (Remarks to the Author):

Thank you for the thoughtful responses and corresponding edits. All of my comments have been adequately addressed.

Reviewer #2 (Remarks to the Author):

Thank you for responding appropriately to the reviewer's questions.

REVIEWERS' COMMENTS

Reviewer #1 (Remarks to the Author):

Thank you for the thoughtful responses and corresponding edits. All of my comments have been adequately addressed.

Author reply: That is good to hear, thank you.

Reviewer #2 (Remarks to the Author):

Thank you for responding appropriately to the reviewer's questions.

Author reply: That is good to hear, thank you.